# Multi-layered proteomic analyses decode compositional and functional effects of cancer mutations on kinase complexes

Martin Mehnert [1 ✉], Rodolfo Ciuffa[1], Fabian Frommelt [1], Federico Uliana[1], Audrey van Drogen[1], Kilian Ruminski[1,3], Matthias Gstaiger[1 ✉] & Ruedi Aebersold [1,2 ✉]

Rapidly increasing availability of genomic data and ensuing identification of disease associated mutations allows for an unbiased insight into genetic drivers of disease development. However, determination of molecular mechanisms by which individual genomic changes affect biochemical processes remains a major challenge. Here, we develop a multilayered proteomic workflow to explore how genetic lesions modulate the proteome and are translated into molecular phenotypes. Using this workflow we determine how expression of a panel of disease-associated mutations in the Dyrk2 protein kinase alter the composition, topology and activity of this kinase complex as well as the phosphoproteomic state of the cell. The data show that altered protein-protein interactions caused by the mutations are associated with topological changes and affected phosphorylation of known cancer driver proteins, thus linking Dyrk2 mutations with cancer-related biochemical processes. Overall, we discover multiple mutation-specific functionally relevant changes, thus highlighting the extensive plasticity of molecular responses to genetic lesions.

[1] Department of Biology, Institute of Molecular Systems Biology, ETH Zurich, Switzerland. [2] Faculty of Science, University of Zurich, Zurich, Switzerland. [3]Present address: Centre d'Immunologie de Marseille-Luminy, Aix Marseille Université, INSERM, CNRS, Marseille, France. ✉email: martin.mehnert@hest.ethz.ch; matthias.gstaiger@imsb.biol.ethz.ch; aebersold@imsb.biol.ethz.ch

The emerging paradigm of personalized medicine essentially posits that the molecular makeup of individual patients will guide their optimal treatment. To date, molecular profiles have been largely generated by a range of genomic technologies, including genome-wide association studies (GWAS), whole-genome sequencing, or transcript profiles. GWAS have established a plethora of correlations between sequence polymorphisms (SNPs) and a variety of human diseases[1]. However, while our understanding of the genetic bases of diseases has progressed at an extraordinary pace, only limited progress has been achieved in systematically mapping the functional repercussions of specific lesions.

Over the last years, mass spectrometry (MS)-based proteomic methods have significantly advanced with respect to their robustness, performance, and scope[2]. With time, methods have been developed, which measure functionally important properties of proteins such as the modification state[3] as well as the composition and topology of protein complexes[4] in addition to the presence and abundance of a protein in a sample. In combination, these MS techniques constitute a powerful tool set to measure mechanistically relevant responses of the cell to (genomic) perturbations.

Protein kinases modulate key cellular processes, form stable and transient complexes with other cellular proteins and are frequently found mutated in genetically linked human diseases. As a consequence kinases are the largest family of current anti cancer drug targets. The CMGC kinase Dyrk2 (dual-specificity tyrosine-phosphorylation-regulated kinase 2) represents one of the less studied kinases and presents several important features relevant for this study.

First, Dyrk2 forms a well-defined multifunctional protein complex encompassing the ubiquitin ligase Ubr5 (also called EDD) and the substrate receptor subunit composed of DDB1 and VprBP (Dyrk2–EDVP [EDD–DDB1–VPRBP complex]). Dyrk2-dependent phosphorylation of recruited substrates is required for the subsequent ubiquitylation by Ubr5 and their proteasomal degradation[5]. Second, Dyrk2 possibly contributes to cancer progression; it has been implicated as both a putative tumor suppressor and an oncogene[6–9]. Third, high-resolution structural information about Dyrk2 is available[10], which facilitates the topological analysis of the kinase module by MS.

In this study, we develop an integrated, multi-layered proteomic workflow to assess the response of cells to specific cancer-associated mutations at various layers of proteomic information. Specifically, we use (i) AP–MS and BioID–MS to assess the effects of specific mutations on the interaction landscape around the mutated protein, (ii) quantitative cross-linking MS to determine alterations in the topology of the complex containing the mutated protein and (iii) quantitative proteome and phosphoproteome profiling by SWATH–MS to quantify the response of proteome and phosphoproteome to the mutated protein. This multi-layered proteomic analysis of the effects of a panel of five Dyrk2 mutants indicates that the cells react to the different mutations with substantial and mutation-specific responses at different levels. We provide several examples of affected functional modules and how the combination of data from different layers may support specific mechanistic explanations. Most prominently, we focus on the discovery of the nuclear pore complex as a cellular module that is affected by Dyrk2 mutations, as well as on how these mutations significantly perturb the interaction with and the phosphorylation of a number of known cancer driver proteins (Cancer Gene Census)[11].

Overall, we describe a multi-layer analysis strategy to determine the molecular response of cells to different cancer-associated mutations in the same gene. The results highlight the complexities of the cellular response to perturbations and provide insights how specific mutations in the Dyrk2 kinase shape cancer-relevant biochemical pathways. The established workflow is generically applicable and its use will aid our understanding of how cancer-associated mutations cause their phenotypic effects.

## Results

**Selection of cancer mutations in the Dyrk2 kinase.** In the first layer of the multi-layered workflow (Fig. 1a), we set out to identify and express a panel of functionally/structurally relevant Dyrk2 mutants. We used as a reference set the 119 mutations in the *DYRK2* gene reported in the COSMIC database (11.08.2015)[12], most of which are missense mutations ($n = 72$) located in the kinase domain (Supplementary Table 4). To select a suitable subset for this study, we considered the frequency of the mutations occurring in *DYRK2* and a damage probability score provided by the structure-ppi algorithm[13], which predicts the impact of missense mutations on proteins based on known structural information, the position of mutations in conserved and functionally relevant regions and the presence of mutations in tumor samples. Overall, we selected four missense (P198L (PL), R378L (RL), S471L (SL), S471P (SP)), and 1 nonsense (S471X (SX)) mutation for further analysis. They all show an increased damage probability score (structure-ppi score: 2–3). The respective positions in the gene and predicted damage score are shown in Supplementary Fig. 1a. The RL mutation is close to the strongly conserved activation loop (Fig. 1b), it has an elevated ppi score of 3 and the respective arginine position has been found to be recurrently mutated ($n = 3$) in *DYRK2*. Furthermore, sequence alignment with Dyrk family members shows that basic residues (either R or K) are conserved at this position[14]. The three selected mutations on position S471 have been identified in patients with breast cancer (SX) and various cancer types (SL, SP)[6,7,15]. Dyrk2 SX contains a stop codon leading to a C-terminally truncated protein variant and has been shown to abrogate the interaction to the EDVP complex member VprBP[16]. The PL mutation is located outside of the kinase domain but close to the DYRK-homology (DH) box, which is strongly conserved among DYRKs[14]. Among the known mutations in this region PL shows one of the highest damage probability scores (structure-ppi score: 2). Finally, we included a catalytically inactive variant of Dyrk2 (Dyrk2 K251R (KR)), which has not been reported in repositories of cancer mutants yet, into the mutant panel[17]. It serves as a control for the evaluation of functional effects of the selected cancer-associated mutations (Fig. 1b).

The selected point mutations were introduced by site-directed mutagenesis into plasmids encoding a Strep-HA-tagged version of Dyrk2 for AP–MS experiments or a BirA*-tagged Dyrk2 variant for BioID–MS experiments. The Dyrk2-mutant constructs were genomically integrated into T-REx HEK293 cells using the Flp–In recombination system and expressed in an inducible manner by addition of doxycycline. As determined by western blot analysis, the generated Dyrk2 mutants were expressed at levels comparable to the wild-type (wt) Dyrk2 (Supplementary Fig. 1b).

**A reference interaction network of the Dyrk2–EDVP complex.** To our knowledge (i) the protein–protein interactions (PPIs) of all EDVP components have not yet been examined, (ii) no proximal interactome, which sheds light on transient interactors of this complex has been reported, and (iii) the overlap between the interactors reported to date for the four EDVP components is very low ($n = 2$, IID (Integrated Interactions Database)[18], probably due to the heterogeneity of the experimental methods used so far. We therefore initially generated an interaction map of the EDVP subunits as a crucial reference interaction network to

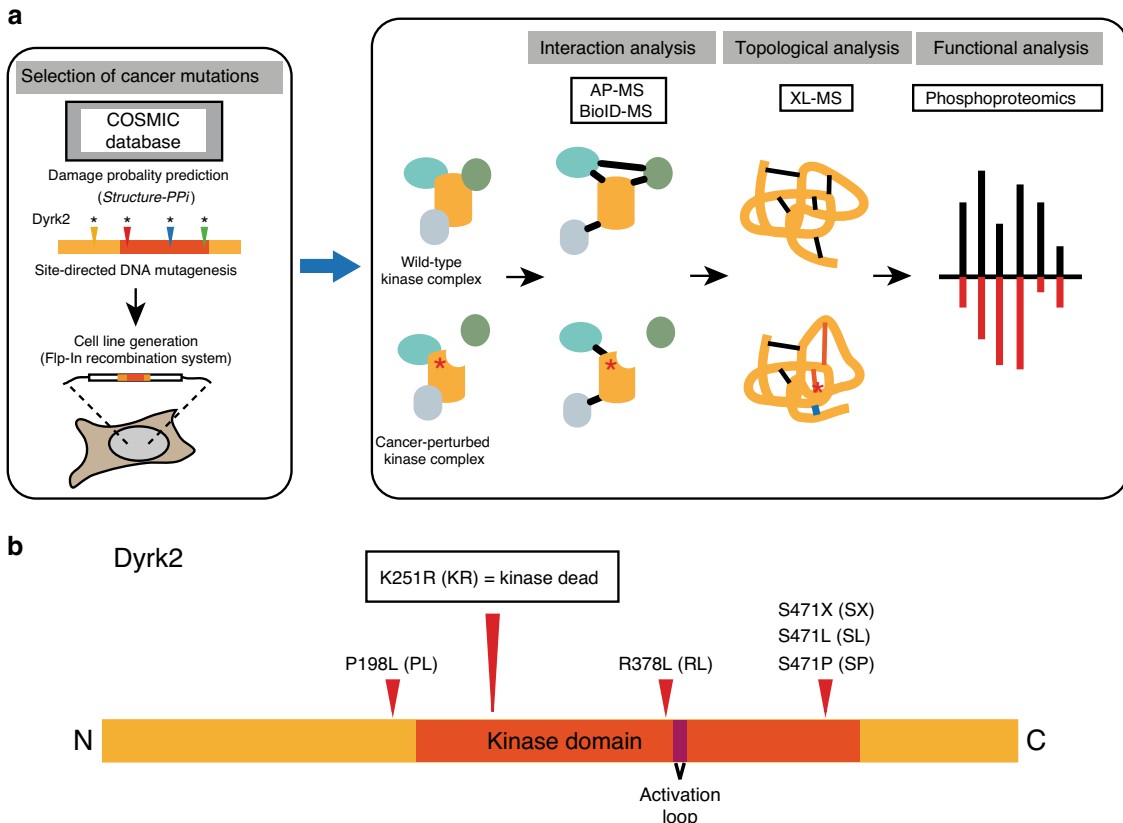

**Fig. 1 Selection of cancer-related mutations for the application of a multi-layered proteomic workflow. a** Applied multi-layered proteomic workflow integrating interaction analysis (AP–MS and BioID–MS), topological analysis (quantitative cross-linking (XL) MS), and functional analysis (phosphoproteomics). **b** Scheme of the position of the selected cancer-related point mutants in Dyrk2.

relate the effects of the chosen Dyrk2 mutations on the core module. To obtain broad coverage of the EDVP interactors and their functional associations, we mapped the interactors of the core module components by two complementary approaches, AP–MS and BioID–MS. Overall, we identified 203 unique high-confidence interactions for all the complex members. The two methods (ranging from 21 PPIs for the ubiquitin ligase Ubr5 to 70 PPIs for Dyrk2) generated highly orthogonal information (average overlap of ~10%). Importantly, the large majority of detected interactions have not yet been reported (on average ~70%). Nine PPIs identified with high confidence were shared between all four EDVP complex members. This represents a more than fourfold increase in identified interactors compared to database knowledge.

Our data both recalled several known EDVP members and identified further proteins associated with the core complex. Among the most extensively characterized interaction modules, we confirmed the interaction of Dyrk2 with the EDVP complex components DDB1, VprBP, and Ubr5 by both methods (Fig. 2, cytoscape graph and lower pie charts). Similarly, the interaction between DDB1 and the E3 CRL4 ligase machinery including several DCAF substrate receptors as well as the known association with the COP9 signalosome was represented in our data set[19–21]. Furthermore, we found that Dyrk2 is prominently associated with members of the Set1A/COMPASS complex and several kinesins, which is in line with a putative role of Dyrk2 in cytokinesis and mitosis[5]. One of the most remarkable findings was the identification of the entire nuclear Y-complex (NUP133, NUP107, NUP85, NUP160, SEH1L, NUP96) in spatial proximity to Dyrk2 by BioID–MS. The Y-complex is a major subunit of the nuclear pore complex involved in nuclear transport processes[22].

Accordingly, functional GO-term analysis showed that in addition to the already suggested involvement of Dyrk2 in cytokinesis and DNA damage repair[5,17,23], it was also strongly linked to various cellular functions related to the nuclear pore complex (Fig. 3a; Supplementary Fig. 2a). In addition, several Dyrk2 interaction partners ($n = 8$) detected in this study such as p53 or the ubiquitin ligase Birc6 are annotated as cancer driver proteins (CDP) in the Cancer Gene Census catalogue[11,24]. Remarkably, the fraction (11.4%) of CDPs is enriched ($p$-value = 0.06) in the Dyrk2 interaction network compared to the overall measured proteome in this study (Fig. 3b).

In combination, AP–MS and BioID–MS confirmed the composition of the Dyrk2 kinase core complex and provided an extended interaction network, which is enriched in cancer driver proteins and processes, as well as other functions.

**Cancer mutations affect the Dyrk2 interaction network.** Next, we monitored the effects of the selected mutations on the wt Dyrk2 interaction network. For this, we used T-REx HEK293 cell lines expressing Strep/HA or BirA*-tagged mutant variants of Dyrk2 and repeated the interaction analysis using AP–MS and BioID–MS. The quantification of protein abundances in inter-action experiments was based on precursor MS1 intensities obtained from MaxQuant v1.5.2.8[25]. Interaction partners were quantified in triplicates with a replicate CV < 21 % (AP–MS) and CV < 10 % (BioID–MS), respectively (Supplementary Fig. 3a).

We first asked to what extent the different mutations affected the kinase core module (Dyrk2–EDVP complex). Previous literature findings[16], damage probability scores and the critical role of catalytic activity suggested that SX, KR, and RL would

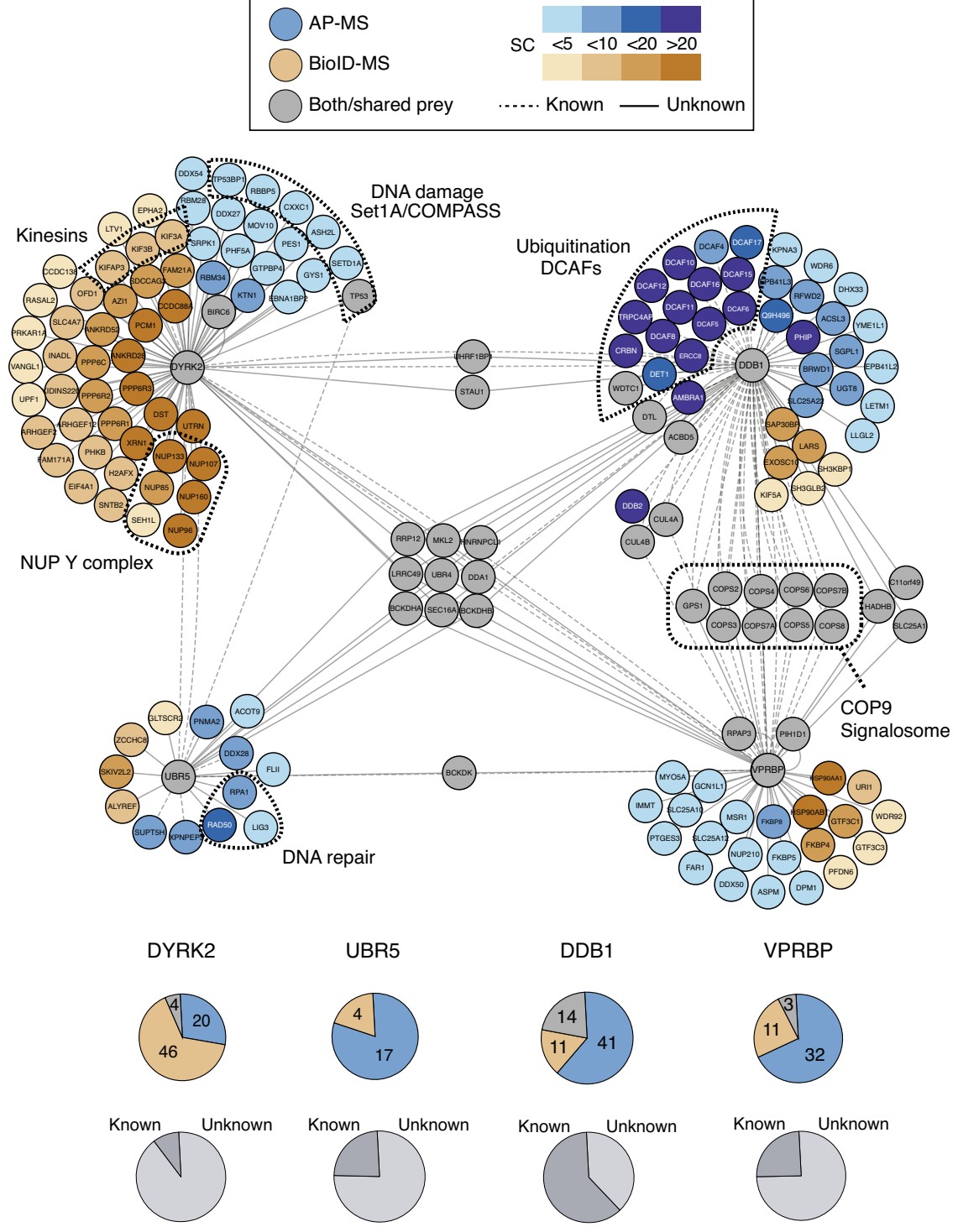

**Fig. 2 Comprehensive interaction network of the wild-type Dyrk2–EDVP complex.** Interaction network of the Dyrk2–EDVP complex. Interactors identified by AP–MS are shown in blue, interactors identified by BioID–MS are shown in brown, interactors identified by both methods are shown in gray. The intensity of the color indicates the number of spectral counts that was assigned to the respective interaction partner. The upper pie charts indicate the number of interactors for the respective bait protein identified by AP–MS or BioID–MS. The lower pie charts show the distribution of known and previously unknown interactors, respectively.

have the greatest impact on the interaction network of Dyrk2 and the assembly of the EDVP complex. This is indeed what we found. Both AP–MS and BioID–MS analyses showed that the truncated Dyrk2-mutant (Dyrk2 SX), the non-cancer-related, catalytically inactive mutant (Dyrk2 KR) and the recurrently mutated Dyrk2 RL, which bears a mutation close to the activation loop, resulted in the most pronounced dissociation from the EDVP complex subunits (Fig. 4a). It is noteworthy that the expression level of the EDVP complex components remained unchanged in the Dyrk2 mutants (Supplementary Fig. 3c). The

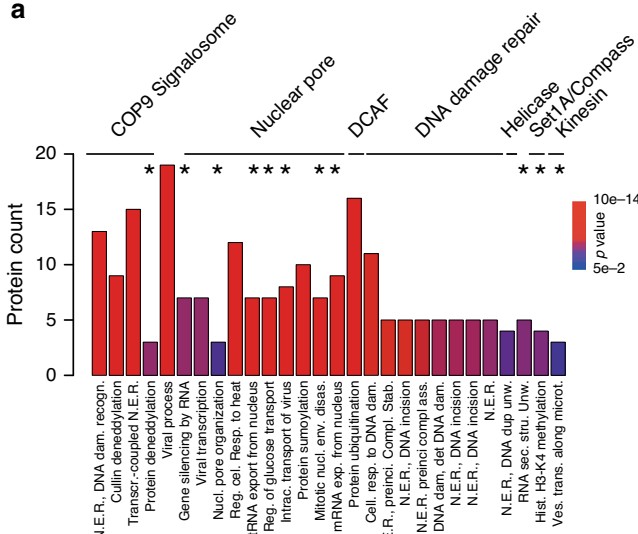

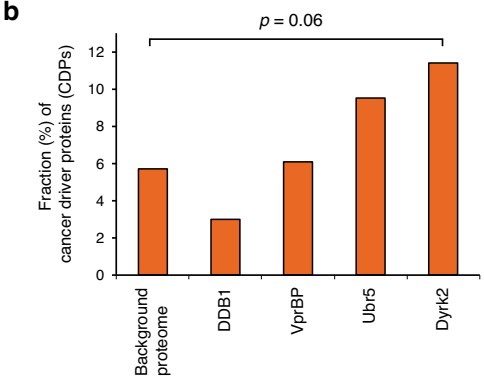

**Fig. 3 GO-term analysis of the Dyrk2–EDVP interaction network. a** Biological processes GO terms enriched (*p*-value ≤ 0.05) for interactors of the Dyrk2–EDVP complex. The asterisk indicates biological processes that were not associated with components of the Dyrk2–EDVP complex yet based on previous interactome data (IID); Nuclear Excision Repair (N.E.R.). The *p*-value for GO-term enrichment (by a modified Fisher exact test) was obtained from DAVID[32]. **b** Fraction of known cancer driver proteins (CDP) (Cancer Gene Census Catalogue) identified in the interactome of the Dyrk2–EDVP complex components and in the measured proteome background. The *p*-value calculation was performed by Fisher's exact test (two-sided). Source data are provided as source data file.

detected differences are therefore caused by changes in protein–protein associations rather than differential protein abundance. In conclusion, we found that certain cancer-related point mutations and the non-cancer-related catalytically inactive Dyrk2-mutant cause a disassembly of the kinase core module revealed by the both complementary interaction approaches AP–MS and BioID–MS.

We next measured the impact of the tested mutations on the entire interactome of Dyrk2. We found that all mutations contributed, albeit to a different degree, to a remodeling of the Dyrk2 protein interaction network (Fig. 4b, c). The extent of interactome rewiring by the mutants broadly reflects the described impact on the association with the EDVP complex. Dyrk2 SX (66 interaction changes with $|\log2FC| > 1$ and adj. *p*-value ≤ 0.05) and the non-cancer-related, catalytically inactive kinase mutant (Dyrk2 KR, 21 interaction changes with $|\log2FC| > 1$ and adj. *p*-value ≤ 0.05) exhibited the greatest impact on the network. They caused a severe to complete

reduction in the number and extent of their interactions. Dyrk2 RL (10 interaction changes with $|\log2FC| > 1$ and adj. *p*-value ≤ 0.05) showed the same patterns of downregulated interactions as the catalytically inactive mutant, although at reduced magnitude (Fig. 4c). In contrast, Dyrk2 PL, SL, and SP showed milder but detectable effects on the interaction network (1–7 interaction changes with $|\log2FC| > 1$ and adj. *p*-value ≤ 0.05). The effect of mutations on single protein modules and the kinase specific PPI network is also reflected by interaction changes involving CDPs. Accordingly, the SX mutant produced the strongest impact on the interaction with CDPs of the Dyrk2 interactome, followed by Dyrk2 RL; on the other hand, the KR mutant, which has not been associated with cancer, showed a rather weak effect contrary to its sizable effect on the global Dyrk2 interactome as a whole (Fig. 4d). The fact that certain point mutations in Dyrk2 substantially disrupt the binding to CDPs supports the putative importance of these mutations in cancer-related processes.

Intriguingly, BioID–MS also indicated that both Dyrk2 KR and Dyrk2 SX severely disrupted Dyrk2 binding to the subunits of the nuclear Y-complex, which we identified in this study as interaction partner of wt Dyrk2 (Fig. 4e). Protein quantification by SWATH–MS revealed no significant change of the expression level of the Y-complex components in the cells expressing Dyrk2 mutants, indicating that the interaction loss is not due to changes in their abundance (Supplementary Fig. 3d). We further performed immunofluorescence microscopic analysis of GFP-tagged wt Dyrk2 and its mutant variants. This demonstrated a clear nuclear translocation defect of Dyrk2 SX, resulting in an exclusive cytoplasmic localization of this mutant. The nuclear localization of Dyrk2 KR, on the other hand, was not affected (Supplementary Fig. 3e). This finding suggests that the Y-complex-related interaction phenotype of Dyrk2 KR is associated with the loss of catalytic activity of Dyrk2 and not with a perturbed cellular localization indicating that specific phosphorylation events at the nuclear envelope could be required for the interaction.

Overall, our data show significant and mutation-specific reorganization of the Dyrk2 interactome upon genetic perturbation of the kinase and a clear effect of some Dyrk2 mutants on the interactions with known CDPs. Intriguingly, the kinase mutant with no association to cancer showed an attenuated effect on the interaction with CDPs. The most significant changes—in particular the disassembly of the EDVP complex and the loss of interactions with the nuclear Y-complex—are associated with mutants that interfere with the catalytic activity of the kinase, suggesting that the kinase activity is required for its binding to key interaction partners.

**Cancer mutations induce topological changes in Dyrk2.** It can be expected that the mutation-dependent changes observed in the Dyrk2 protein interaction network were the result of changes in Dyrk2 structure or function. We therefore determined the Dyrk2 phosphorylation patterns and topological reorganization of Dyrk2 in the mutant panel.

Enriched Strep/HA-tagged Dyrk2-mutant variants from AP–MS pulldown experiments were analyzed for changes in their state of phosphorylation within the Dyrk2 sequence. Bait-normalized abundance values from MaxQuant v1.5.2.8 were used for the quantification of the identified Dyrk2 phosphopeptides. Overall, we found that the different mutants affected the phosphorylation state of Dyrk2 substantially and to different extents. Furthermore, in general the decrease in Dyrk2 phosphorylation level (Fig. 5a) correlated well with the impact of the respective mutant on the interactome (Fig. 4b). In particular,

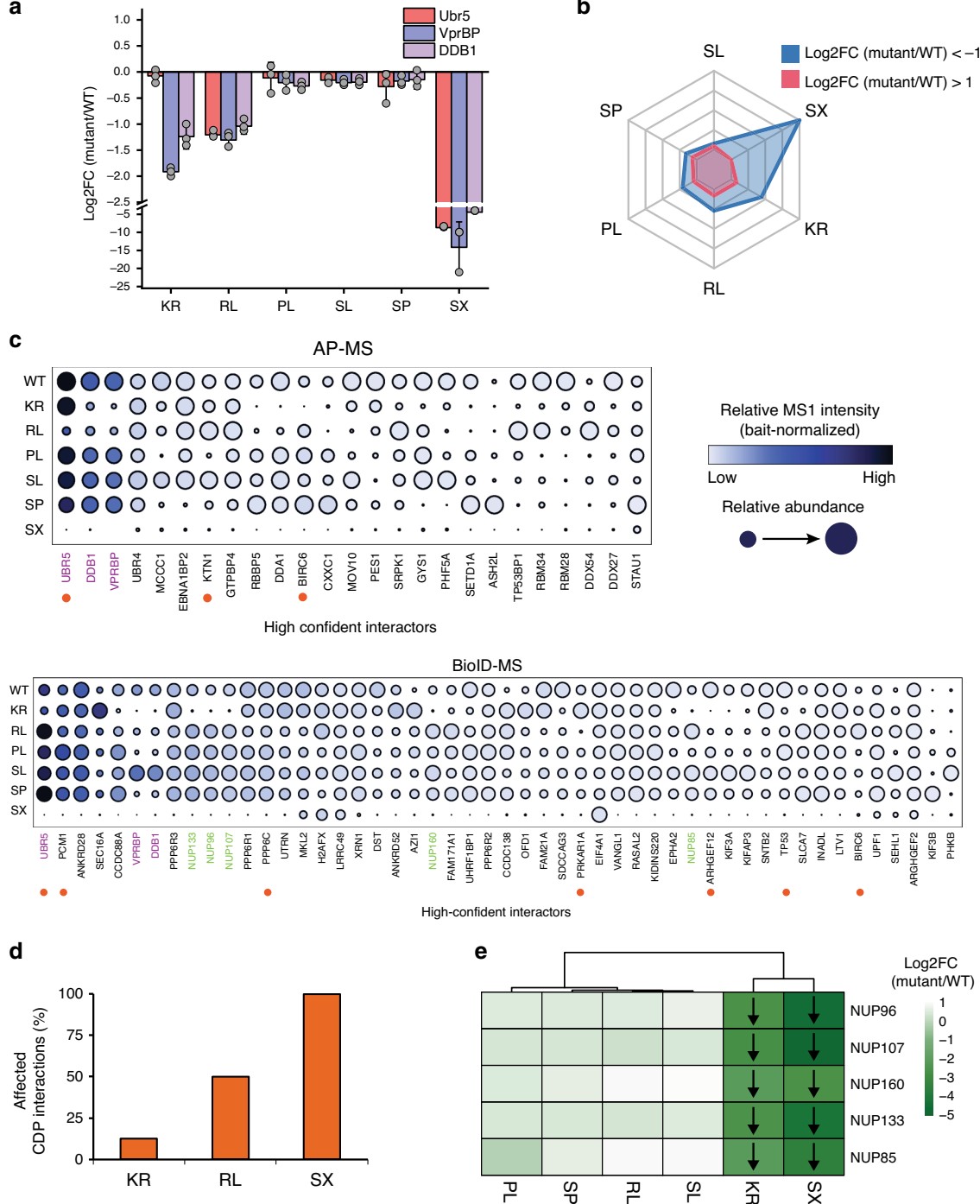

**Fig. 4 Cancer-related point mutations perturb the Dyrk2 interactome. a** Effect of cancer-related mutations on the interaction to the core subunits of the Dyrk2–EDVP complex measured by AP–MS. Error bars denote the mean value (center) with 95% confidence interval ($n = 3$ biologically independent experiments). **b** Radar chart representing the number of changed interactors (|log2FC| >1) measured by AP–MS and BioID–MS. **c** Mutation perturbed Dyrk2 interactome measured by AP–MS (upper panel) or BioID–MS (lower panel). The color indicates the bait (Dyrk2) normalized MS1 intensity. The node size displays the relative abundance of an interactor across the mutant conditions. Dyrk2–EDVP subunits are highlighted in purple letters, nuclear Y-complex subunits are highlighted in green letters. Cancer driver proteins (CDPs) are marked with a red dot. **d** Effect of Dyrk2 mutants on the interaction with cancer driver proteins (CDPs) (|log2FC| >1, adj. $p$-value ≤ 0.05). Dyrk2 SL, SP, and PL do not affect CDP interactions. **e** Certain point mutations affect the binding of Dyrk2 to the Y-complex of the nuclear pore. The arrow indicates the direction of a significant change in binding (adj. $p$-value ≤ 0.05). Statistical analysis was performed with two-tailed unpaired Student's $t$-test and $p$-values were adjusted using the method of Benjamini–Hochberg[67]. Source data are provided as source data file.

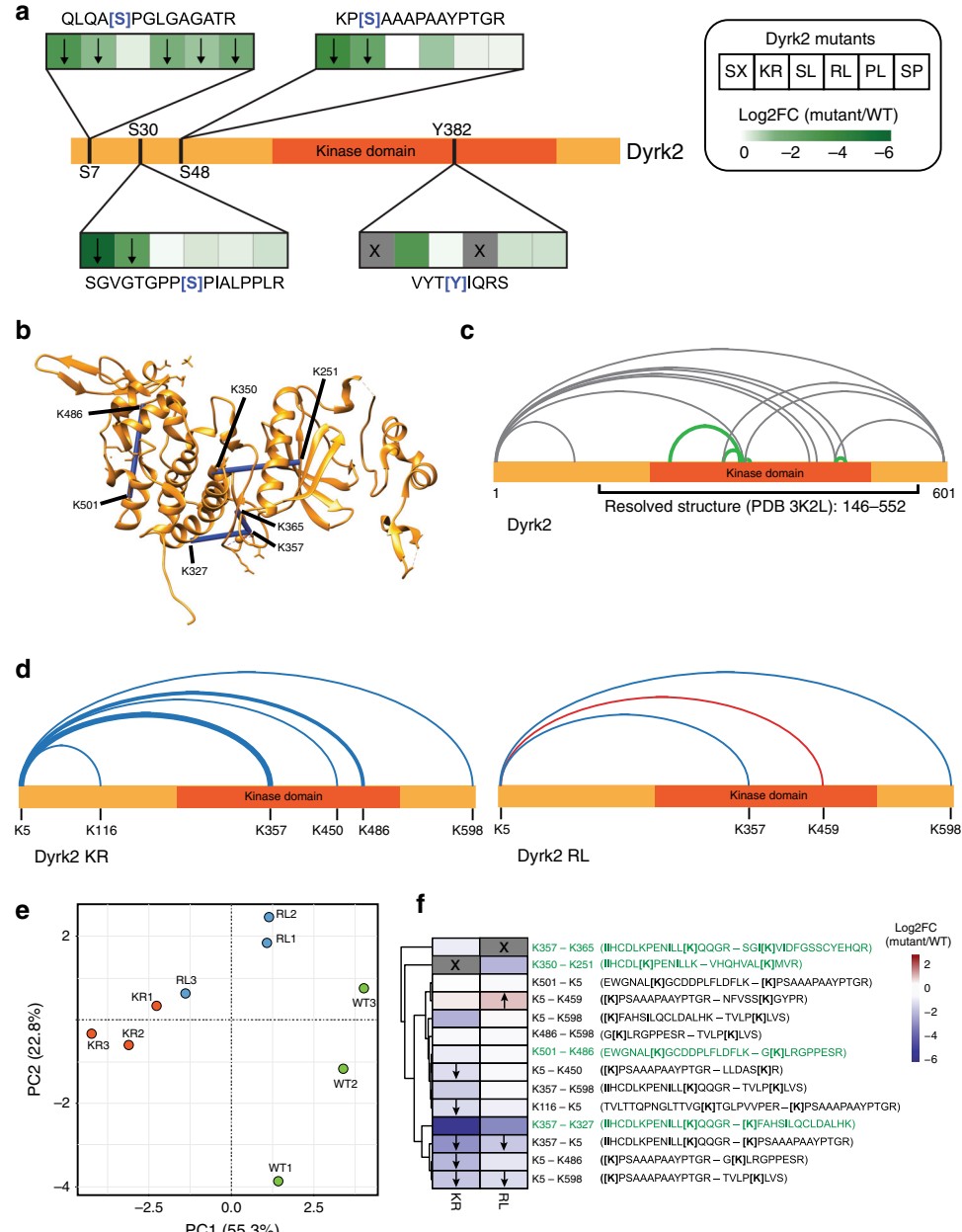

**Fig. 5 Differential topological analysis of cancer-related Dyrk2 mutants by cross-linking MS. a** Effect of cancer-related mutations on the phosphorylation status of Dyrk2 measured by AP–MS. The arrow indicates the direction of a significant change in phosphorylation ($p$-value ≤ 0.05). Statistical analysis was performed with two-tailed unpaired Student's $t$-test. The cross represents conditions in which no peptide could be detected. **b** Mapping of intraprotein cross-links that satisfy the distance threshold (~30 Å) of the cross-linker DSS to the structure of Dyrk2 (PDB: 3K2L). The Dyrk2 structure comprises the region between the amino acids 146–552 and lacks the N- and C-terminal part of the protein. **c** Visualization of confidently identified intraprotein cross-links (ld score ≥ 25). In green are shown cross-links that are validated by the known structure of Dyrk2. Gray lines represent cross-links that are located in regions for which structural information is missing. **d** Visualization of cross-links affected by selected point mutations in Dyrk2. Blue and red lines indicate a significant downregulation (blue)- or upregulation (red) of the cross-link (adj. $p$-value ≤ 0.05). Statistical analysis was performed with two-tailed unpaired Student's $t$-test and $p$-values were adjusted using the method of Benjamini–Hochberg[67]. The thickness of the line stands for the degree of the change. Cross-links that were not affected by the mutations are not shown. **e** Principal component analysis (PCA) of quantified cross-linked peptides between wt Dyrk2 and its mutant variants. **f** Heatmap showing the effect of selected Dyrk2 point mutations on the abundance of cross-linked peptides. The arrow indicates the direction of a significant change in peptide abundance (adj. $p$-value ≤ 0.05). Statistical analysis was performed as in **d**. The cross indicates cross-linked peptides that could not be detected due to a mutation of the cross-linking residue (KR mutant) or the cleavage site (RL mutant). Cross-linked lysine residues are shown in brackets and structurally validated cross-linked peptides are highlighted in green. Source data are provided as source data file.

catalytically inactive Dyrk2 KR, Dyrk2 SX and to a lesser extent Dyrk2 RL clearly reduced phosphorylation, including the autophosphorylation of the activation loop (Y382), which is crucial for the topology and activation of the kinase[10,26]. Of note, the mutation of arginine to leucine in Dyrk2 RL blocks the tryptic

cleavage at this position preventing the identification of the peptide comprising the activation loop sequence. For this reason, the effect of Dyrk2 RL on the phosphorylation of the activation loop is still unclear. Together, the differential Dyrk2 phosphorylation state may lead to topological changes in the

mutants, which in turn has an impact on the Dyrk2 interaction fitness.

To identify topological changes accompanying differential phosphorylation and interaction patterns, we combined chemical cross-linking (XL) with targeted mass spectrometry (PRM) (see "Methods" section). We selected the mutants that revealed the strongest effect on the interaction network: the catalytically inactive KR mutant as well as the cancer-associated RL and SX mutants. Wild-type Dyrk2 and the selected mutants were recombinantly expressed in SF9 insect cells and purified using a FLAG-tag. Dyrk2 SX was also expressed but could not be purified from SF9 cells, possibly due to a misfolding of this protein under these conditions (Supplementary Fig. 4a, b). The purified Dyrk2 variants were treated with the lysine-specific cross-linker DSS followed by an enrichment of trypsin-digested cross-linked peptides by size-exclusion chromatography (SEC; Supplementary Fig. 4c). The differential quantitative cross-linking analysis was carried out by PRM using a precursor library containing cross-linked peptides of the different mutant conditions identified by the xQuest/xProphet software pipeline (ld (linear discriminant)—score > 25; Supplementary Fig. 4d)[27]. We quantified 14 cross-links across the different mutant conditions. Cross-linking experiments with affinity-purified Strep/HA-tagged wt Dyrk2 matched to 8 of the 14 cross-links (Supplementary Table 3). In the correlation analysis using normalized transition intensity, all replicates emerged as individual clusters with high Pearson correlation values (average $R = 0.91$ (KR), 0.95 (RL), 0.90 (WT)) (Supplementary Fig. 4e, f). Furthermore, principal component analysis (Fig. 5e) and fold-change calculation (Fig. 5f) revealed measurable topological changes across the different mutants, each displaying a specific pattern and magnitude of change. The identified cross-links can be broadly subdivided in two groups: four of the identified cross-links, highlighted in green in the primary sequence map of wt Dyrk2 (Fig. 5c), form the first group and are exclusively positioned within the kinase domain. These cross-links could be mapped to the known crystal structure of Dyrk2 (PDB: 3K2L) and all fell within the expected DSS distance restraint of ~30 Å (Fig. 5b)[10,28], validating the correctness of the cross-linking procedure. The second group comprises the remaining ten high-confidence cross-links mapped to the N-terminal or C-terminal region of the kinase, for which no structural information is currently available (Fig. 5c). This group of cross-links suggests that the potentially disordered N-terminal region of the protein (IUPRED; data not shown) is making contact with large portions of the protein, primarily the C-terminal region of the kinase domain. In both mutants the first group of cross-links remained largely unaltered, indicating that the structure of the kinase domain was not fundamentally reorganized. In contrast, in both mutants the interaction of the N-terminal region with the kinase domain was perturbed: in the Dyrk2 KR mutant the interaction was strongly decreased (e.g., K357-K5, log2FC −2.8, adj. $p$-value = 0.02). In contrast, the Dyrk2 RL mutant showed a mixed pattern, including increased interaction between the N-terminal region and the C-terminal lobe of the kinase domain and decreased interaction between the N terminus and the rest of the protein (Fig. 5d, f).

In conclusion, our topological data are in good agreement with prior structural information and define specific topological changes associated with the individual mutants. The abundance differences of cross-linked peptides quantified by the PRM method supported the comparison of topological changes. In absence of additional experimental data, we can only speculate about the exact relationship between the here described phosphorylation and topological changes and the protein interaction network. It is, however, reasonable to hypothesize that some of the changes captured in our analysis can rationalize the already discussed rewiring in the interactome of Dyrk2 mutants.

**Effect of Dyrk2 mutants on the cellular phosphoproteome.** The wild-type form of the Dyrk2 kinase is expected to phosphorylate a specific subset of the cellular proteome. We hypothesized that the mutation-induced changes in the topology and phosphorylation state of Dyrk2, and the ensuing changes in the core module and the extended PPI network would leave a detectable footprint in the cellular phosphoproteome and thus provide important functional insights about the cellular functions affected by the mutation. We therefore performed phosphoproteomic and proteome abundance analyses of cells expressing Dyrk2 point mutations. Dyrk2 mutants were genomically integrated via the Flp–In recombination system into a Dyrk2 KO cell line (T-REx-HeLa) engineered by CRISPR/Cas9 (Supplementary Fig. 5a, b) and expressed in an inducible manner by the addition of doxycycline (Supplementary Fig. 5c). The deletion of endogenous *DYRK2* in the cell lines prevented a dilution of the mutant-specific phosphoproteomic phenotypes due to residual Dyrk2 wild-type activity, thus increasing the sensitivity of the phospho-phenotyping. The quantitative phosphoproteomic analysis was performed by SWATH–MS, followed by phosphosite determination using LuciPHOr2[29,30] (Supplementary Note 1).

Hierarchical clustering of the phosphopeptide patterns revealed a common pattern of phosphoproteome dysregulation for the catalytically inactive Dyrk2 cell lines (KO and KR) and the C-terminally truncated Dyrk2-mutant (Dyrk2 SX) (Fig. 6a). Specifically, in these cell lines, we observed a significant number of downregulated phosphopeptides compared to cells expressing wt Dyrk2. Between 21% and 47% of the quantified phosphopeptides (adj. $p$-value ≤ 0.05) were downregulated with a log2 fold-change ≤ −1 in the kinase mutants (KR: 47%, 68 peptides, SX: 21%, 54 peptides) and the Dyrk2 KO (33%, 48 peptides). For Dyrk2 SX, we also detected a fraction (8%, 20 peptides) of upregulated phosphopeptides (Fig. 6c).

Next, we analyzed the sequences of phosphopeptides with decreased abundance and found that these were enriched in the phosphorylation motif recognized by the Dyrk2 kinase (R/Kxx(x)S/TP)[31] (Fig. 6d). This suggests that many of these phosphosites represent putative direct substrates of Dyrk2 (e.g., MISP, log2FC (KO) = −2.2; MEP50, log2FC (KO) = −2.1; Supplementary Table 1). In keeping with this finding, mutations that showed the most significant phosphoproteomic footprint also most severely affected the kinase interaction network and generated a strong interaction phenotype, as shown in the previous layer of our proteomic workflow (Fig. 4b, c). The second main cluster in our phosphoproteomic analysis contains mutations with mild interaction phenotype (Dyrk2 PL, SL, SP) that also showed only weak effect on the phosphoproteome. To further confirm the observed phosphoproteomic patterns, we performed in vitro kinase assays with recombinant Dyrk2 variants (see "Methods" section, in vitro ADP-Glo kinase assay). The results confirm that with the exception of the Dyrk2 KR mutant the tested kinase mutants were still catalytically active and show comparable activity to wt Dyrk2 (Supplementary Fig. 6a). Surprisingly, we found that the expression of Dyrk2 PL, SL, and SP caused an upregulation of several phosphopeptides (Fig. 6c). For Dyrk2 PL, the highest fraction (21% related to all quantified phosphopeptides, adj. $p$-value ≤ 0.05) of upregulated phosphopeptides were found. Here, 20% of the upregulated phosphopeptides showed an abundance change of log2FC > 2. Notably, the upregulated phosphosites did not contain the Dyrk2 target motif or any other clearly enriched phosphorylation motif, implying that the

enhanced phosphorylation rather resulted from indirect effects of the Dyrk2 mutants on these proteins (Supplementary Fig. 6b).

Besides changes in Dyrk2 catalytic activity the observed phosphoproteomic phenotype could be also due to protein abundance changes of the phosphorylated proteins or Dyrk2 itself or finally to a combination of these effects. To rule out that the observed effects on phosphopeptide level were due to protein abundance changes, we performed a total proteome analysis by

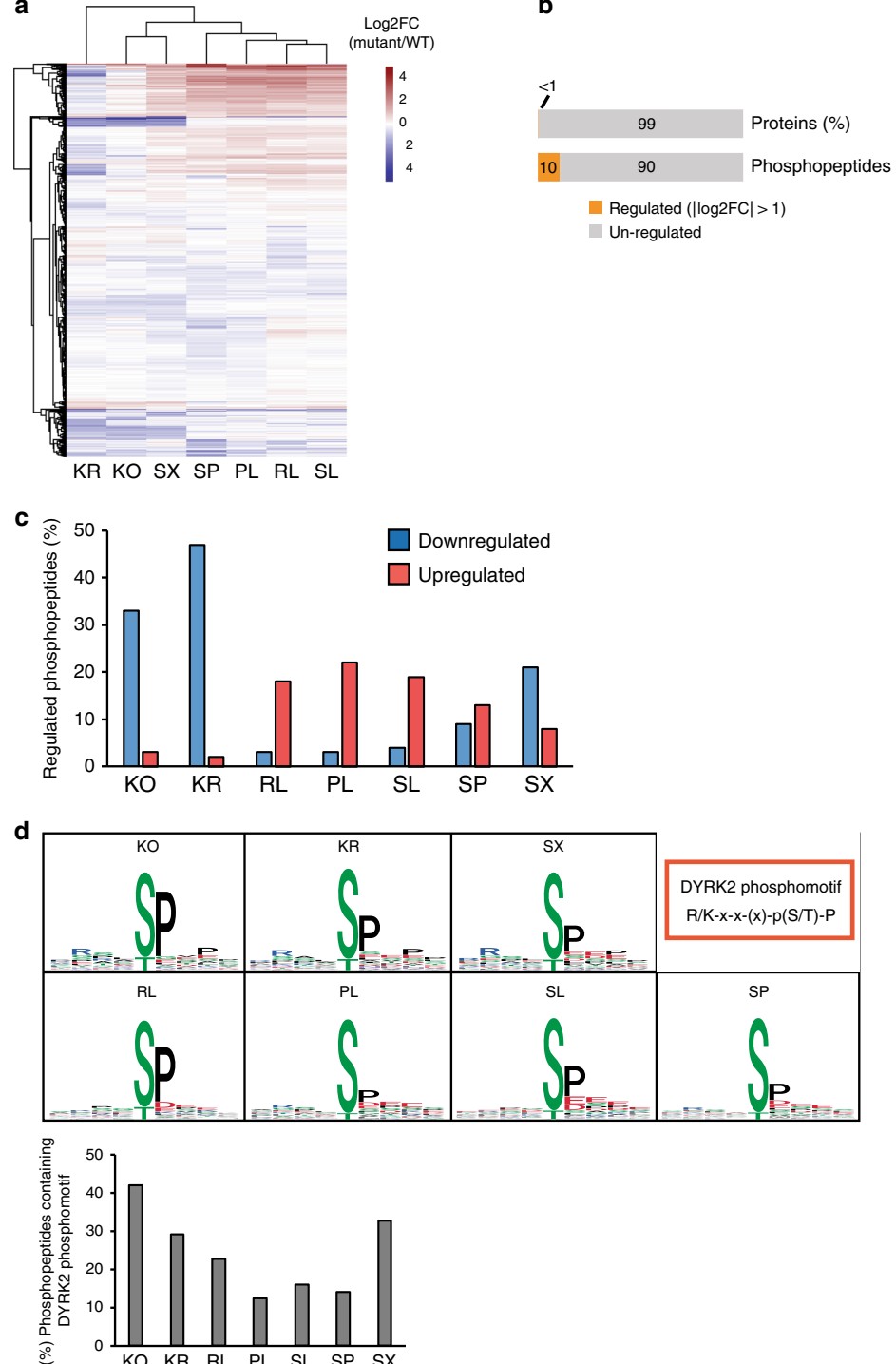

**Fig. 6 The molecular footprint of Dyrk2 cancer-related mutations on the cellular phosphoproteome. a** Heatmap with hierarchical clustering illustrating changes in the abundance of cellular phosphopeptides across the different cancer-related Dyrk2 point mutants. Only those peptides that have been detected at least once as statistically significantly changed (adj. *p*-value ≤ 0.01, │log2FC│ > 0.5) are displayed. **b** Content of regulated proteins and phosphopeptides summarized from all Dyrk2-mutant cell lines (regulated phosphopeptides: │log2FC│ > 1, adj. *p*-value ≤ 0.05). **c** Fraction of significantly upregulated or downregulated localized phosphopeptides in cell lines expressing the Dyrk2 point mutants (│log2FC│ > 1, adj. *p*-value ≤ 0.05). **d** Phosphomotif analysis of downregulated phosphopeptides in the different Dyrk2 variants (adj. *p*-value ≤ 0.05) (left panel). Fraction of downregulated phosphopeptides containing the Dyrk2 phosphomotif (right panel). The statistical analysis and *p*-value calculation (adj. *p*-value (FDR)) was performed within the mapDIA package[60]. Source data are provided as source data file.

SWATH–MS, and consistently measured and quantified 5138 proteins (protein CV < 11%; protein FDR ≤ 3%) (Supplementary Fig. 5g) across the cell lines. Compared to the phosphoproteomic readout, we found only mild changes in protein levels upon Dyrk2-mutant expression, indicating that protein abundance measurement is less informative than the other analyzed layers for the cellular impact of mutations. Across all mutant conditions, only 0.2% (10 proteins, adj. $p$-value ≤ 0.05, $|\log2FC| > 1$) of proteins showed altered abundance and, in contrast to the phosphoproteome analysis, the catalytically inactive and truncated Dyrk2 mutants did not emerge as cluster (Figs. 6b and 7a). Instead, Dyrk2 RL and SL clustered together and showed the highest number of significantly regulated protein abundances (Dyrk2 RL: 107 proteins, Dyrk2 SL: 94 proteins, adj. $p$-value ≤ 0.05, $|\log2FC| > 0.5$). However, nearly all these proteins are only moderately regulated, with a log2FC < 1 (RL: 103 proteins, SL: 94 proteins) indicating that the mutants generated a rather minor and unspecific protein abundance footprint. Furthermore, we found that (i) in the Dyrk2 KO as well as in the other mutant cell lines the altered phosphopeptide abundances did not correlate with the corresponding protein abundances ($R^2 = 0.0199$, Dyrk2 KO) (Fig. 7b; Supplementary Fig. 6c); (ii) the abundance of the Dyrk2 mutants did not change compared to wt Dyrk2, except for Dyrk2 SX that shows an elevated protein level (Supplementary Fig. 5c). Taken together, these results suggest that changes on phosphoproteome level result from genuine differential phosphorylation events induced by point mutations or the deletion of *DYRK2*.

To identify cellular processes that might be affected by the Dyrk2 mutations on phosphoproteome level, we performed an enrichment analysis of GO terms on significantly regulated phosphosites (adj. $p$-value ≤ 0.05, $|\log2FC| > 1$) from our data set using DAVID (https://david.ncifcrf.gov/)[32] (Fig. 7c; "Methods" section).

In all mutants, we observed a regulation of phosphosites belonging to proteins annotated to function in "mRNA processing" (GO: 0006397) and "Cell-cell-adhesion" (GO: 0098609). For the latter, we found a significant enrichment in the inactive (Dyrk2 KR, $p$-value = 2.05e−04) as well as in the putatively still active Dyrk2 PL mutant ($p$-value = 2.56e−06), consistent with the reported role of Dyrk2 in cancer metastasis[9,33,34]. Certain GO terms were enriched only in a specific mutant group. In the catalytically inactive (Dyrk2 KO, KR) and truncated mutant (Dyrk2 SX), a significant number of regulated phosphosites belonged to proteins involved in nuclear transport processes (e.g., "RNA export from the nucleus" (GO: 0006405, $p$-value = 2.5e−02 (KR), $p$-value = 3.10e−05(KO))), consistent with the disrupted binding of these mutants to the nuclear pore complex observed in our differential interaction analysis. Remarkably, both the serine missense mutations (Dyrk2 SL and Dyrk2 SP) lead to a high number and significant enrichment (Dyrk2 SL, $p$-value = 3.38e−05; SP, $p$-value = 1.55e−04) of regulated phosphosites linked to transcriptional control ("positive regulation of transcription from RNA polymerase II promoter" (GO:0045944)) in line with the involvement of Dyrk2 in transcriptional processes and the regulation of protein synthesis[35–37].

Interestingly, we also detected 150 phosphopeptides (80 localized phosphopeptides, FLR ≤ 1%) belonging to CDPs. Among them, 23 phosphosites were significantly regulated ($|\log2FC| > 1$, adj. $p$-value ≤ 0.05) upon mutation or deletion of *DYRK2*. Consistent with its impact on CDP interactions, Dyrk2 SX caused the strongest effect on the phosphorylation of CDPs, together with another cancer-related Dyrk2 point mutant (Dyrk2 PL) (Fig. 7d). In keeping with its effect on the interaction with CDPs, the non-cancer-related Dyrk2 KR mutant showed a rather minor effect on the phosphorylation of CDPs.

Interestingly, some of the regulated CDP phosphosites were found to be mutated in cancer tissues (e.g., NPM1 S125, PL (log2FC = 2.2), RL (log2FC = 2.04), SP (log2FC = 2.58)[38]) or are known to have a functional role in protein–protein interactions (e.g., HMGA1 T53, Dyrk2 KR (log2FC = 1.05), PL (log2FC = 1.21), and SX (log2FC = 1.14))[39,40] (Supplementary Table 2).

Together, our data reveal mutation-specific phosphoproteomic signatures that can be both used to discriminate between different mutant backgrounds and to gain insights into their phenotypic effects. We observed a strong downregulation of several phosphopeptides in those mutants having the strongest effects on the interactome of Dyrk2, involving in particular processes affected also in the interaction data, such as nuclear transport.

**Network of cancer driver proteins perturbed by Dyrk2 mutants**. Finally, we asked whether our data, combined with prior knowledge about cancer proteins and PPI's, could help to identify important regulated modules and thereby rationalize the putative role of the selected mutants in cancer.

The interaction analysis described above revealed that Dyrk2 is associated with a higher-than-random number of CDPs (Fig. 3b). Furthermore, we found a clear effect of some cancer-related Dyrk2 mutations on the interaction with CDPs (Fig. 4d). To understand the interplay between interaction and phosphoproteome CDP regulation, we combined the relevant data in a PPI network based on previously deposited interactions between regulated proteins ("Methods" section; Supplementary Note 2) (Fig. 8a). We found that the CDPs identified in our interaction and phosphoproteome analysis form a network in which, overall, Dyrk2 SX and RL exhibit the strongest regulatory effect (Figs. 7d and 8a). Functional GO-term analysis of the regulated CDPs revealed an enrichment of biological processes such as "negative regulation of cell proliferation" (GO:0008285, $p$-value = 7.3e−4), "negative regulation of apoptotic processes" (GO:0043066, $p$ = 1.4e−3) (Supplementary Note 3), "chromatin remodeling" (GO:0006338, $p$-value = 5.3e−4), and "nuclear transport" (GO:0051169, $p$-value = 9.2e-3). Indeed, the deletion of *DYRK2* significantly elevated proliferation in T-REx-HeLa cells as shown by colony formation and MTT assay (Fig. 8b; Supplementary Fig. 7a). The result is in line with previous xenograft mouse studies[8] and supports a putative tumor suppressor function of Dyrk2.

The phosphorylation and interaction of CDPs involved in "nuclear transport" were mainly affected by the catalytically inactive Dyrk2 mutants (Dyrk2 KO, KR) and Dyrk2 SX, in keeping with their strong effect on the global phosphoproteome (Fig. 8a; Supplementary Note 4). Out of 18 localized NUP phosphosites (FLR ≤ 1%) that were measured in our phosphoproteomic analysis, only sites of NUP214, NUP98, and NUP88 were regulated by the Dyrk2 mutants KR and SX as well as Dyrk2 KO (Supplementary Fig. 7c). Interestingly, NUP214 and NUP88 are direct interaction partners and are associated with the nuclear Y-complex that was identified as interactor of Dyrk2 in this study[41,42]. As shown above, the expression of Dyrk2 KR and SX perturbed the binding to the subunits of the nuclear Y-complex, which is in line with the significantly reduced phosphorylation of the nearby subunits NUP214 and NUP88 in these mutants identifying the nuclear pore complex as cellular module that is affected at different functional levels (interaction and phosphorylation) by Dyrk2 mutations.

In order to elucidate a putative functional role of Dyrk2 on the nuclear pore complex (NPC), we aimed at validating the Dyrk2-dependent phosphorylation of the cancer driver protein NUP214, which is suggested to act as docking site in nuclear transport processes[43]. Indeed, in vitro phosphorylation of NUP214 with

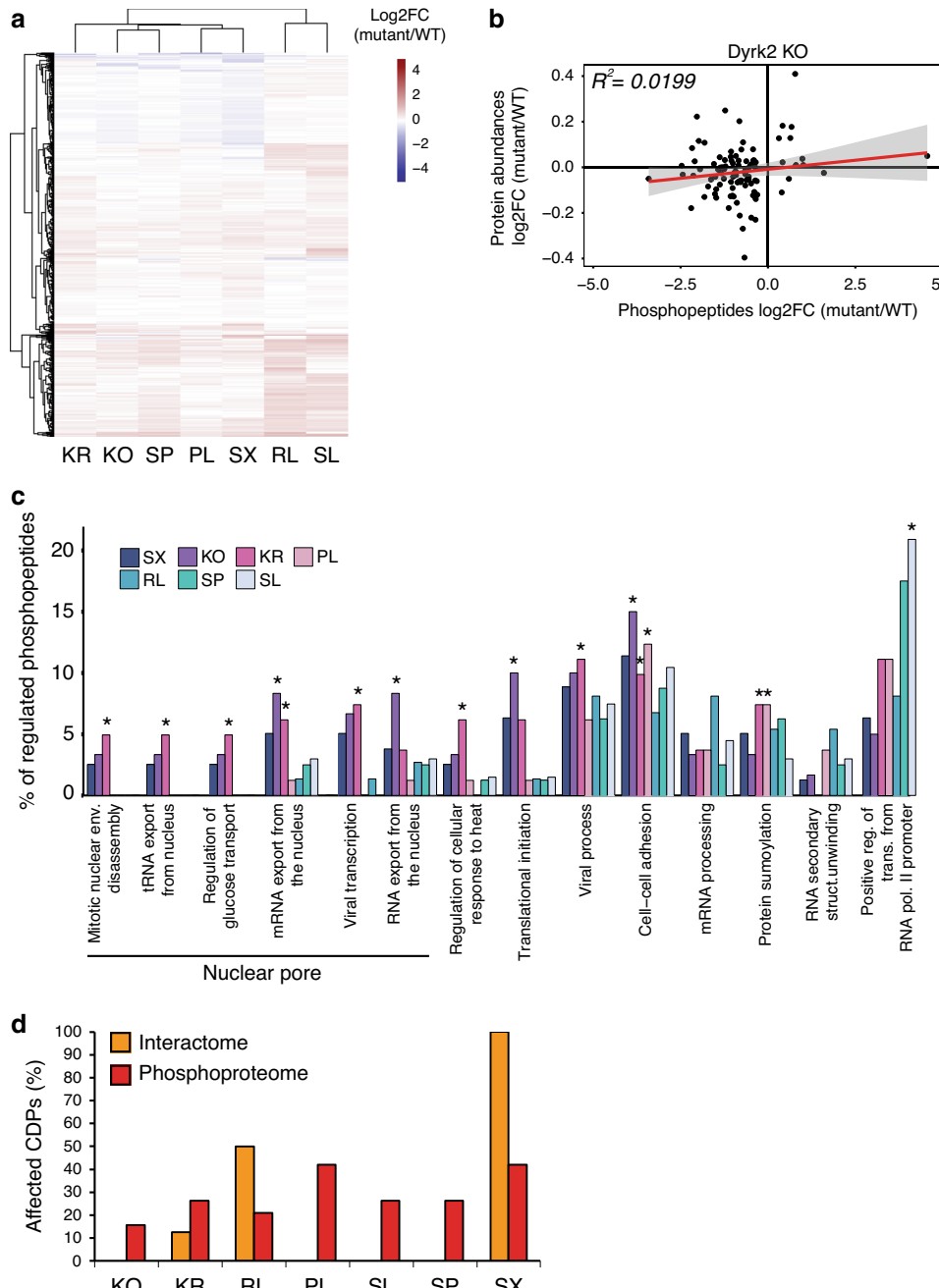

**Fig. 7 Further proteomic and phosphoproteomic analysis of Dyrk2 mutants. a** Heatmap showing differences in protein abundances across the different cancer-related Dyrk2 point mutants. Only those proteins that have been detected at least once as statistically significantly changed (adj. $p$-value $\leq 0.01$, $|\log2FC| > 0.5$) are displayed. **b** Correlation of log2 fold changes between significantly regulated phosphopeptides and the corresponding protein identified in *DYRK2* KO cells (adj. $p$-value $\leq 0.05$). **c** Biological processes GO-term analysis of regulated phosphopeptides identified in the different Dyrk2 mutants. The asterisk indicates an significant enrichment of the GO term ($p$-value $\leq 0.05$). The $p$-value for GO-term enrichment was obtained from DAVID[32]. **d** Effect of Dyrk2 point mutations on the interaction and phosphorylation of cancer driver proteins identified in the interaction and phosphoproteome analysis of this study ($|\log2FC| > 1$, adj. $p$-value $\leq 0.05$). The statistical analysis and $p$-value calculation (adj. $p$-value (FDR)) was performed within the mapDIA package[60]. Source data are provided as source data file.

recombinant Dyrk2 confirmed the protein as direct phosphorylation substrate of Dyrk2 (Fig. 8d). Notably, the phosphorylation state of NUPs has been shown to be important for their binding to the NPC and the regulation of NPC assembly[44], which also influences other NPC-related functions such as nuclear transport.

Based on this data, we propose a model for the putative role of Dyrk2 at the NPC where the interaction with the nuclear Y-complex promotes the phosphorylation of nearby NUPs by Dyrk2 (Fig. 8e), which potentially influences the association of these NUPs with the NPC.

Overall, our data illustrate how the combination of different layers of proteomic information not only improves our ability to discriminate between the effects of different mutants and interpret their functional and cellular effects, but also suggests mechanistic and causal links exemplified by the impact of Dyrk2 mutants on components of the nuclear pore complex. We

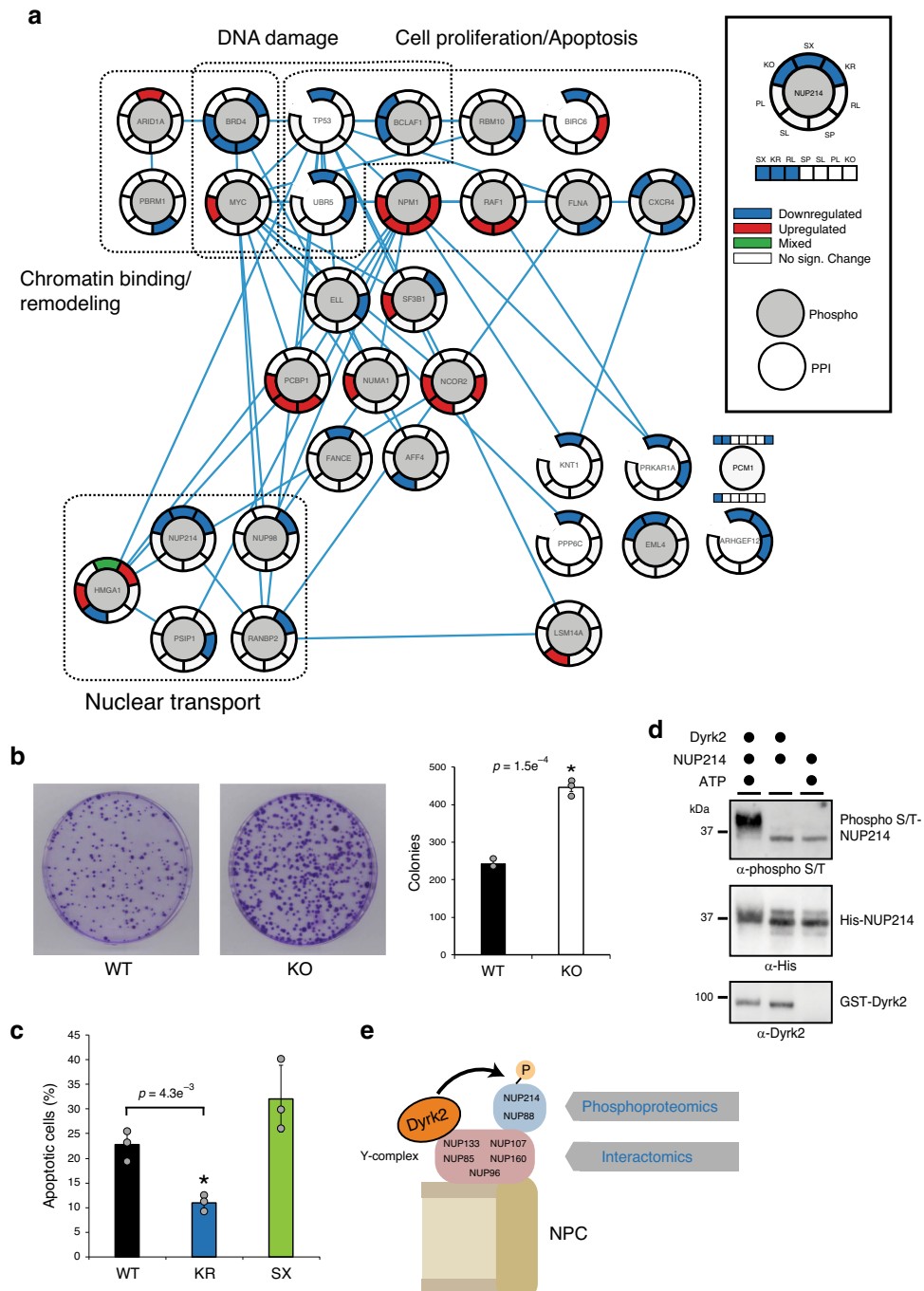

**Fig. 8 Impact of Dyrk2 mutations on the network of known cancer driver proteins. a** Network of cancer driver proteins (Cancer Gene Census) found to be significantly regulated either at the phospho level (|log2FC | > 1, adj. *p*-value ≤ 0.05) and/or at the interactome level (|log2FC | > 1, adj. *p*-value < 0.05). The statistical analysis and *p*-value calculation (adj. *p*-value (FDR)) was either performed within the mapDIA package (phosphodata) or with two-tailed unpaired Student's *t*-test and *p*-values were adjusted using the method of Benjamini–Hochberg[67] (interaction data). The network is based on reported interactions between the proteins (IID v.2018-05; only experimentally validated interactions). The gene ontology analysis was performed with DAVID and significant terms (*p*-values < 0.05) were merged in general terms ("Methods" section). The regulation of the interaction and phosphorylation of cancer driver proteins by Dyrk2 mutants is illustrated with a color code. For PCM1 the lower color code shows the regulation on interactome level, the upper color code shows the regulation on phosphorylation level. **b** Colony formation assay of CRISPR/Cas9 engineered T-REx-HeLa *DYRK2* KO cells. **c** Annexin V-FITC apoptosis assay of MDA-MB-213 cells overexpressing Dyrk2 wt or Dyrk2 KR and Dyrk2 SX, respectively. **d** In vitro phosphorylation assay using recombinant Dyrk2 and NUP214 (Ser601-Arg868) incubated for 1 h at 37 °C. Phosphorylation was detected by an anti-phospho-serine/threonine antibody. The experiment was repeated independently (*n* = 2) with similar results. **e** Model of the putative function of Dyrk2 at the nuclear pore complex (NPC). As shown by BioID–MS Dyrk2 interacts with subunits of the nuclear Y-complex. Quantitative phosphoproteomic analysis revealed a Dyrk2-dependent phosphorylation of NUP214 and NUP88, which represent direct interaction partners of the Y-complex. Error bars represent the mean value (center) with 95% confidence interval (*n* = 3 biologically independent experiments). Statistical analysis was performed with two-tailed unpaired Student's *t*-test. The asterisk indicates a significant statistical difference (*p*-value ≤ 0.05). Source data are provided as source data file.

identified Dyrk2 SX and Dyrk2 RL as cancer-associated mutants with the strongest impact on the phosphorylation and interaction to CDPs, suggesting a contribution of these mutants, probably together with other factors, in cancer progression.

## Discussion

In recent years, high-throughput DNA sequencing technologies have been used to generate very large cancer genomic data sets indicating the degree of genomic variation in different cancer types. Many of the discovered variants have then been statistically associated with cancer[6,7]. However, at present it is impossible to predict from genomic data alone whether and how specific genomic mutations cause molecular changes that eventually lead to pathological phenotypes. In the absence of this information, algorithms that predict a damage probability score and thus the impact of point mutations on proteins have been developed and used[13,45,46]. The damage score considers the position in functionally relevant or conserved domains affected by the mutations and their presence in tumor tissue. Although the calculation of these scores partially relies on prior knowledge like interaction interfaces, phosphorylation sites, and 3D models, they do not necessarily reflect the real functional impact of a certain point mutation on the protein. In addition, classical damage probability scores only predict the effect of the mutation on a single protein but do not consider the consequences on the entire cellular system. In this respect, there is a fundamental lack of streamlined procedures to globally evaluate the molecular consequences of clinically relevant mutations.

In this study, we set out to directly measure the effects of cancer-related mutations in the *DYRK2* kinase gene on the composition, function and topology of the Dyrk2 kinase complex, the extended Dyrk2 PPIs and the cellular phosphoproteome, thereby providing a comprehensive view of the repercussions of cancer mutations at different levels of cellular systems. To address this goal, we integrated different proteomic techniques into a multi-layered workflow that allowed us to quantify the effects of cancer-associated mutations at different proteomic levels. In contrast to increasingly widespread strategies devised to combine different omics layers, most frequently genomic, transcriptomic, and proteomic data, our workflow is not based exclusively on global abundance values but explores other, functionally directly relevant proteomic information. The strategy is based on the assumption that the combination of information about the changes induced by the respective mutants at the level of protein modules, extended PPI's, the functional state and the functional effect of the mutated protein on the proteome, can more directly pinpoint mechanisms propagating the effects of single mutations to the entire cellular system than can be presently predicted[47].

For this study, we chose the CMGC kinase Dyrk2 as prototypical kinase module for the development and application of this proteomic workflow because of its central role in the assembly of a multi-subunit protein complex and its putative involvement in cancer progression[48]. The analyzed point mutations were selected based on their presence in tumor tissues, the calculated damage probability score and their frequency of observation in the Dyrk2 kinase.

Our proteomic workflow showed that despite the similar damage probability score the different cancer-related mutations lead to different cellular responses on the interactome and phosphoproteome level. Catalytically inactive (Dyrk2 KR) and the C-terminal truncated (Dyrk2 SX) mutant caused a strong perturbation of the interaction network and lead to a disassembly of the Dyrk2 kinase core complex, eventually resulting in a perturbed proteolytic degradation of substrates, which in turn may affect cellular processes. In this context, it is worth noting that, as

previously shown[49], AP–MS and BioID-MS prove largely complementary and should be utilized in combination to obtain an in-depth mapping of the molecular context. The same mutations also disrupt the binding to the nuclear Y-complex, which was identified as interactor of Dyrk2 in this study, and affect the phosphorylation status of Dyrk2. The phosphorylation of specific residues in kinases has been shown to drive and change their protein topology and the catalytic activity[50]. In fact, quantitative cross-linking MS revealed significant differences in the protein topology between the mutants indicating that the observed interaction phenotypes underlie topological changes in the kinase. Our integrated proteomic workflow further showed that mutations affecting the interaction network also lead to significant changes in the cellular phosphoproteome. Here, we found that different mutations affect phosphosites belonging to proteins involved in different cellular processes suggesting that each mutant creates its own molecular (phospho) footprint. A severe malfunction of the kinase complex is reflected by perturbations on several proteomic layers such as the interactome and phosphoproteome, which ultimately results in cellular phenotypes as shown for Dyrk2 KR and Dyrk2 SX. However, also for mutations that affect the kinase interaction network only moderately, we observed changes in the phosphoproteome. We therefore classify the mutations into two groups: (i) mutations with severe effect on several proteomic layers (composition, function and topology) leading to a malfunction of the kinase complex (Dyrk2 KR, SX); and (ii) mutations with a mild phenotypic effect (Dyrk2 RL, PL, SL, SP) that may not necessarily result in a malfunction of the kinase complex. Interestingly, for class I mutants, we found that to some extent the same cellular modules in the interaction and phosphoproteome analysis were affected. This observation is exemplified by the effect of Dyrk2 mutations on the nuclear pore complex. Beside the association with the nuclear Y-complex Dyrk2 KR and Dyrk2 SX also affect the phosphorylation of nuclear pore subunits, such as NUP214, NUP88, and NUP98 (Supplementary Discussion).

Importantly, our analysis further revealed that the selected cancer-related mutations have a significant impact on the interaction with and the phosphorylation of known CDPs annotated in the Cancer Gene Census catalogue[11,24]. As expected from the global interactome and phosphoproteome analysis, Dyrk2 SX exhibits the strongest regulatory effect on CDPs followed by Dyrk2 RL. Although its overall phenotype is rather weak, Dyrk2 RL seems to specifically affect CDPs. The RL point mutation is closely positioned to the activation loop of Dyrk2 that is crucial for the activity and topology of the kinase[10,26], and may thus explain the observed phenotypic effects of this mutant. In conclusion, the integrated proteomic workflow identified Dyrk2 SX and Dyrk2 RL as mutants with the strongest impact on the phosphorylation and interaction to CDPs suggesting a putative involvement of these mutants, possibly together with other factors, in cellular processes leading to cancer progression. It is remarkable that single point mutations produce a measurable effect on different proteomic layers as shown by our workflow. The nature and magnitude of such changes is at present hardly predictable from prior knowledge, and need to be determined experimentally. A combination of subtle molecular phenotypes derived from various point mutations in different protein complexes may ultimately lead to pathological phenotypes.

Overall, in this study, we developed and applied a multi-layered proteomic workflow for a system-wide phenotypic characterization of cancer mutations in kinase complexes. We could show that mutations assigned with a similar damage probability score or kinase activity affect protein topologies, posttranslational modifications, interaction network, and cellular systems to a different extent. In particular, we observed clear differences

between the mutants regarding their effect on the interaction with and the phospho network of CDPs, which might be crucial for the role of protein mutations in cancer processes. In contrast, in vitro kinase assays of the different Dyrk2 mutants were unable to reflect the phenotypic differences detected by the proteomic workflow, demonstrating that determination of kinase activity alone is not sufficient to discriminate mutations with similar damage score. In this respect, our workflow has a much higher discriminative power and may provide a better damage assessment of point mutations than conventional damage probability scores or kinase activity assays. Thus, the output of such integrated proteomic approach also may help to discriminate potential cancer driver mutations from passenger mutations. Classical molecular biology and biochemical approaches usually focus on single proteins or protein complexes, which precludes the investigation of mutations on large scale and in a system-wide manner. In contrast, this study shows that the combination of different proteomic techniques serves as powerful tool to study the functional consequences of genomic mutations on a broad scale (Supplementary Discussion).

## Methods

**Plasmids and cloning**. For the generation of expression constructs encoding a N- or C-terminal twin-Strep and hemagglutinin (SH)-tag (pTO-SH) for affinity purification experiments or a N- or C-terminal FLAG-BirA*-tag[51] for BioID experiments entry clones of a Gateway compatible clone collection (ORFeome v5.1) were used. The integration of the entry clones into the Gateway destination vectors was performed with an enzymatic LR clonase reaction (Invitrogen). Site-directed mutagenesis of Dyrk2 constructs was performed using Pfu Ultra High Fidelity DNA-polymerase according to the manufacturer's instructions (Agilent Technologies). For the generation of the MultiBac vector expressing FLAG-tagged Dyrk2 (pFBDM-FLAG-Dyrk2) in SF9 cells, the coding sequence of Dyrk2 was amplified by PCR using oligonucleotides encoding the FLAG-tag and subcloned into pDNOR221 with BP clonase reaction (Invitrogen) (Supplementary Table 5). The DNA region encoding FLAG-Dyrk2 was inserted into pFBDM using NotI and HindIII restriction sites. The bacmid for the transfection of SF9 cells was generated by heat shock transformation of DH10Bac E. coli cells (#10361012, Invitrogen) with 2 µg pFBDM-FLAG-Dyrk2 and recovery at 37 °C for 8 h. Afterwards, cells were spread on agar plates containing Ampicillin, Kanamycin, Tetracycline, and Gentamycin. Grown colonies were used for the preparation of the bacmid DNA.

**Tissue culture and DNA transfection**. T-REx-HEK293 Flp–In (#R78007, Invitrogen) and T-REx-HeLa (#R71407, Invitrogen) cell lines were cultured in DMEM (4.5 g/l glucose, 2 mM L-glutamin) (Gibco) supplemented with 10% fetal bovine serum (BioConcept), 100 U/ml penicillin (Gibco), and 100 µg/ml streptomycin (Gibco) at 37 °C in a humidified incubator with 5% CO$_2$. For DNA transfection, cells were treated with jetPrime (Polyplus) according to the manufacturer's instructions.

**Cell line generation**. T-REx Flp–In cells were co-transfected with the respective expression plasmid and the pOG44 vector (Invitrogen) encoding the Flp-recombinase. Two days after the transfection, the selection of cells undergone recombination was initiated by addition of 15 µg/ml blasticidin C and 100 µg/ml hygromycin to the media for 2–3 weeks.

**CRISPR/Cas9-mediated gene knock-out**. CRISPR-guided RNAs (MM051/052 and MM063/064; Supplementary Table 5) were designed based on their specificity score retrieved from the Optimized CRISPR Design web tool (http://crispr.mit.edu) (gRNA 1 target sequence: 5′-ACCGGGGAGAAAACGTCAGT-3′, gRNA 2 target sequence: 5′-GGACAGCATTCATAGACGGC-3′). Annealed DNA oligonucleotides containing the target sequence were cloned into the hSpCas9 plasmid (pX458, Addgene) using BbsI restriction sites. T-REx-HeLa (#R71407, Invitrogen) cells were transfected with two hspCas9 constructs encoding gRNAs that target the third exon of the target gene DYRK2. The cell culture medium was replaced 4 h after transfection and cells were recovered for 72 h. For FACS sorting, 1 × 10e6 cells were gently detached from the tissue culture plate with 0.25% trypsin-EDTA (Gibco) and resuspended in PBS containing 1% FBS. GFP-positive cells were isolated by FACS (BD Facs Aria IIIu sorter) and single cells were sorted into each well of a 96-well plate. Cell clones were expanded for 3 weeks and then screened for deletion events by western blotting.

**Western blot**. Western blot analysis was performed with cells lysed in 100 µl lysis buffer (0.5% NP40, 50 mM Tris–HCl, pH 8.0, 150 mM NaCl, 50 mM NaF supplemented with 1 mM PMSF and protease inhibitors (Sigma)). The cell lysate was

cleared by centrifugation (15,000×g, 20 min) and boiled for 5 min after addition of 3× Laemmli sample buffer. The denatured sample was loaded on NuPAGE 4–12% Bis-Tris SDS-PAGE gels (Invitrogen) for gel electrophoresis and then transferred onto nitrocellulose membranes (Trans-Blot Turbo, BioRad). The following primary antibodies were used: anti-Dyrk2 (HPA027230, Sigma; 1:200), anti-DDB1 (D4C8, #6998, Cell Signaling; 1:1000), anti-Ubr5 (D608Z, #65344, Cell Signaling; 1:1000), anti-VprBP (#14966, D5K5V, Cell Signaling; 1:1000), anti-FLAG (F3165, Sigma; 1:5000), anti-HA (HA.11,901513, BioLegend; 1:5000), and anti-Actin (ab179467, Abcam; 1:1000). Proteins were detected by enhanced chemiluminescence (ECL, Amersham) using horseradish-peroxidase-coupled secondary antibodies (#7074, Cell Signaling (1:5000) and #115035003, Jackson ImmunoResearch (1:5000)). Uncropped images of the western blots are shown in the Source Data file.

**Immunofluorescence**. T-REx-HeLa cells (#R71407, Invitrogen) expressing GFP-tagged Dyrk2 variants were fixed on coverslips for 10 min with 600 µl 4 % Paraformaldehyde. After washing with PBS fixed cells were permeabilized with 0.2 % Triton-X 100 for 5 min, washed with PBS and then incubated with 600 µl 2% BSA for 30 min. Nuclear staining was performed with Hoechst (1:10,000, 10 mg/ml stock solution). Coverslips were mounted on the microscope slide with ProLong Gold antifade reagent (Invitrogen).

For the analysis of localization of Dyrk2 WT, KO, and SX mutant, more than 20 cells per condition were imaged on a wide-field Olympus MM microscope. Images were analyzed in Image J Fiji v1, cytoplasmic and nuclear signals were obtained by manual and automatic segmentation, respectively, and their ratio calculated.

**Colony formation assay**. T-REx-HeLa cells (#R71407, Invitrogen) were washed twice with PBS and then fixed with ice cold 100% methanol for 15 min. After fixation cells were washed with PBS and staining was performed with 5% Crystal violet (Sigma) solution for 20 min at room temperature. Then cells were washed with PBS until the dye was no longer draining out.

**MTT assay for cell proliferation estimation**. T-REx-HeLa cells (#R71407, Invitrogen) were seeded in 6-well plates and treated with 0.1 mg/ml MTT (3-[4,5-dimethylthiazol-2-yl]-2,5-diphenyltetrazolium bromide, Sigma) for 3 h at 37 °C. The medium was removed and the converted dye was solubilized in 100% iso-propanol. The absorbance of the converted dye was measured at a wavelength of 570 nm with background subtraction at 670 nm (Synergy HT, BioTek).

**Cell cycle analysis**. Cell cycle analysis was performed with the Propidium Iodide Flow Cytometry Kit (Abcam, ab139418) according to the manufacturer's instructions. Briefly, T-REx-HeLa cells (#R71407, Invitrogen) were washed with PBS and then incubated with propidium iodide and RNase for 30 min at 37 °C. The stained cells were analyzed with the BD LSRFortessa (BD Biosciences) flow cytometer and data were analyzed with FlowJo v9.7.6.

**Apoptosis assay**. MDA-MB-231 cells (ATCC HTB-26) were analyzed for apoptosis using the Annexin V-FITC Apoptosis Staining/Detection Kit (Abcam, ab14085) according to the manufacturer's instructions. Briefly, cells were harvested and resuspended in binding buffer. Then cells were incubated with Annexin V-FITC and propidium iodide for 5 min at room temperature followed by flow cytometry analysis with BD LSRFortessa (BD Biosciences) flow cytometer. The data were analyzed with FlowJo v9.7.6.

**In vitro phosphorylation of NUP214**. Commercial recombinant NUP214 (Aviva Systems Biology, OPCD05854) was incubated with 100 ng commercial recombinant Dyrk2 (Promega, V5090) in kinase reaction buffer (Promega, V9101) supplemented with ATP (0.1 µM) and DTT (1 mM) for 1 h at 37 °C. The reaction was stopped by adding 3× Laemmli buffer and heating up at 95 °C for 5 min.

**In vitro ADP-Glo kinase assay**. The in vitro kinase assay to estimate the activity of Dyrk2 mutants used in this study was performed with the ADP-Glo kinase assay kit (Promega, V9101) according to the manufacturer's instructions. Briefly, equal amounts of recombinant Dyrk2 variants purified from SF9 insect cells were incubated in kinase reaction buffer (Promega, V9101) together with ATP (10 uM), DTT (50 µM) and 3 µg substrate (DYRKtide) (Promega, V9101) for 1 h at 37 °C. Then ADP-Glo reagent was added to stop the kinase reaction, and to convert remaining ADP to ATP and the sample was incubated for 40 min at room temperature. After that kinase detection, reagent was added followed by incubation for 60 min at room temperature. Luminescence was measured in white 96-well plates (ThermoFisher, 265302) with a plate reader (Synergy HT, BioTek) using 1 s integration time.

**Affinity purification**. The expression of SH-tagged bait proteins stably integrated in T-REx-HEK293 Flp–In cells was induced with 1 µg/ml doxycycline for 24 h. For affinity purification, four confluent 150 mm tissue culture plates were harvested and the cell pellet was snap-frozen. Then the cell pellet was lysed in 4 ml lysis buffer (0.5% NP40, 50 mM HEPES (pH 7.5), 150 mM NaCl, 50 mM NaF, 400 nM

Na3VO4 supplemented with 1 mM PMSF, 1.2 µM Avidin, and protease inhibitor cocktail (P8849, Sigma)). The cleared cell lysate was incubated with Strep-Tactin beads (IBA LifeSciences) for 1 h on a rotation shaker. Upon washing two times with lysis buffer and three times with the same buffer but without protease inhibitor and detergent, the protein complexes were eluted from the Strep-Tactin beads with 2 mM biotin. Proteins of the eluate were precipitated with trichloroacetic acid and then dissolved in 8 M urea. Cysteine bonds were reduced with 5 mM Tris(2-carboxyethyl)phosphine (TCEP) and alkylated with 10 mM iodoacetamide. The proteins were digested with 0.8 µg trypsin (V5112, Promega) over night followed by peptide clean-up with C18 UltraMicroSpin columns (The Nest Group). The dried peptides were dissolved in 2% acetonitrile and 0.1% formic acid.

**BioID**. One subconfluent (80%) 150 mm plate of T-REx-HEK293 Flp–In cells stably expressing FLAG-BirA*-tagged bait proteins was incubated for 24 h with 1 µg/ml tetracycline for protein expression. Then the media was replaced and the cells were incubated with 50 µM biotin for additional 24 h. After collection of the cells and centrifugation (400×g, 5 min), the cell pellet was snap-frozen. Lysis of the cell pellet was performed in 1 ml RIPA buffer (50 mM Tris–HCl (pH 8), 150 mM NaCl, 1% Triton-X 100, 1 mM EDTA, 0.1% SDS supplemented with 1 mM PMSF and protease inhibitor cocktail (Sigma)) followed by Benzonase (Sigma) treatment (250 U) at 10 °C for 30 min. The cleared lysate was then incubated with disuccinimidyl suberate (DSS) (Sigma) cross-linked Strep-Tactin beads (IBA LifeSciences) for 1 h on a rotation shaker. The beads were washed three times with RIPA buffer, three times with HNN buffer (50 mM HEPES (pH 7.5), 150 mM NaCl, 50 mM NaF), and two times with 100 mM NH4CO3. Proteins bound to the beads were denatured with 8 M urea, reduced with 5 mM Tris(2-carboxyethyl)phosphine TCEP and alkylated with 10 mM iodoacetamide. The sample was diluted with 100 mM NH4CO3 to 4 M urea and proteins were digested on the beads with 0.5 µg LysC (Wako) for 3 h followed by dilution to 1 M urea and digestion by 0.8 µg trypsin over night. The digestion was stopped by addition of 5% formic acid and the peptides were purified by C18 UltraMicroSpin columns and dried in a speedvac. The dried peptides were dissolved in 2% acetonitrile and 0.1% formic acid.

**Protein extraction and in-solution digest**. For total proteome analysis T-REx-HeLa cells (#R71407, Invitrogen) were washed with ice cold PBS, scrapped off from the plate and snap-frozen in liquid nitrogen. The cell pellet was lysed in 8 M urea and sonicated three times for 1 min (Hielscher-Ultrasound Technology) followed by centrifugation at 18,000×g for 45 min to remove insoluble material. The protein amount of the cleared supernatant was measured by the Bicinchoninic acid (BCA) assay (Pierce) and 100 µg protein were subsequently reduced with 5 mM Tris(2-carboxyethyl)phosphine TCEP for 30 min at 37 °C and alkylated with 10 mM iodoacetamide for 30 min at 37 °C in the dark. The protein sample was diluted to 4 M urea with 100 mM NH4CO3 and digested by LysC (protease/protein ratio 1:100) for 4 h. The sample was further diluted to 1.5 M urea with 100 mM NH4CO3 and digested with Trypsin (protease/protein ratio 1:75) over night. The digestion was stopped by addition of 5% formic acid and peptides were purified by C18 columns (Sep-Pak, Waters). The desalted peptides were washed with 5% acetonitrile and 0.1% formic acid, eluted with 50% acetonitrile and 0.1% formic acid, and dried in a speedvac. The dried peptides were dissolved in 2% acetonitrile and 0.1% formic acid and iRT peptides (Biognosys) were added.

**Phosphopeptide enrichment**. T-REx-HeLa cells (#R71407, Invitrogen) were washed with ice cold PBS, scrapped off from the plate and snap-frozen in liquid nitrogen. The cell pellet was lysed in 8 M urea and sonicated three times for 1 min (Hielscher-Ultrasound Technology) followed by centrifugation at 18,000×g for 45 min to remove insoluble material. The protein amount of the cleared supernatant was measured by the Bicinchoninic acid (BCA) assay (Pierce) and 500 µg protein were then reduced with 5 mM Tris(2-carboxyethyl)phosphine (TCEP) for 30 min at 37 °C and alkylated with 10 mM iodoacetamide for 30 min at 37 °C in the dark. The protein sample was diluted to 4 M urea with 100 mM NH4CO3 and digested by LysC (1:100) for 4 h. The sample was further diluted to 1.5 M urea with 100 mM NH4CO3 and digested with Trypsin (1:75) over night. The digestion was stopped by addition of 5% formic acid and the peptides were purified by C18 Ultra-MicroSpin columns. The dried peptides were dissolved in loading buffer for the enrichment of phosphopeptides (50% acetonitrile, 0.1% trifluoroacetic acid, 300 mg/ml lactic acid) and incubated with 5 mg TiO2 beads for 30 min at room temperature under strong shaking. The following steps were performed as previously described[52]. Phosphopeptides were eluted with 50 mM (NH4)2HPO4, pH 10.5 and the pH of the eluate was immediately adjusted to pH 2–3 with trifluoroacetic acid. Afterwards the sample was desalted with C18 columns (Sep-Pak, Waters) and dried in a speedvac. The dried phosphopeptides were dissolved in 2% acetonitrile and 0.1% formic acid and iRT peptides (Biognosys) were added.

**Insect cell culture and transfection**. SF9 cells (#11496015, Invitrogen) were cultured in SF9 insect cell media (Gibco) supplemented with PGS (Gibco) in an incubator (IKA shaker) at 27 °C and 300 rpm in the dark. After reaching a density of 2–3 × 10e6 cells/ml, the cells were diluted to 0.7 × 10e6 cells/ml. DNA transfection was carried out with GENEJuice transfection reagent (Sigma) according

manufacturer's instructions. Briefly, 2 × 10e6 cells were seeded in a 6-well plate and treated with 5 µg bacmid DNA added to the transfection reagent. After incubation for 72 h at 27 °C, the cell suspension was centrifuged for 5 min at 3000×g and the cleared supernatant containing the generated baculovirus was used for the infection of SF9 cells.

**Protein expression and purification**. For protein expression, 450 ml SF9 cell suspension culture at a density of 2 × 10e6 cells/ml were infected with baculovirus solution in a ratio 1:10 and incubated at 27 °C and 300 rpm in the dark for 3-4 days. At a cell viability of <70%, the SF9 cells were harvested by centrifugation for 5 min at 3000×g and the pellet was snap-frozen in liquid nitrogen. For protein purification, the cell pellet was lysed in 250 ml lysis buffer (50 mM HEPES, 150 mM NaCl, 50 mM NaF, 5% Glycerol supplemented with 400 nM Na3VO4, 1 mM PMSF, 1 mM Tris(2-carboxyethyl)phosphine (TCEP) and protease inhibitor cocktail (1:500) (Sigma)) and treated with 4000 U Benzonase (Sigma) for 30 min at room temperature. Afterwards, the lysate was sonicated and centrifuged to remove cell debris at 16,000×g for 20 min. The cleared lysate was incubated with FLAG M2 agarose beads (Sigma) for affinity purification of FLAG-tagged Dyrk2 over night on a rotation shaker at 4 °C. Afterwards, the beads were washed with lysis buffer without PMSF and protease inhibitors, and FLAG-tagged Dyrk2 was eluted with 1 mg/ml of FLAG peptide (ApexBio, Houston TX). The eluate was purified by size-exclusion chromatography with Superdex 75 10/300GL (GE Healthcare, Uppsala SWE) with running buffer (50 mM HEPES pH 8.0, 150 mM NaCl) to remove the excess of FLAG peptide. Samples were concentrated with 5 kDa molecular weight cut-off spin column (Vivaspin 500, Sartorius) to a final protein concentration of 0.5 mg/ml determined by Bicinchoninic acid (BCA) assay. The quality of the purification (higher than 90%) was assessed by SDS-PAGE.

**Cross-linking of Dyrk2 purified from SF9 insect cells**. Purified Dyrk2 was cross-linked at a concentration of 0.5 mg/ml with 1 mM isotope labeled di-succinimidylsuberate (DSS-d0, DSS-d12) (CreativeMolecules Inc.) at 37 °C for 30 min as previously described[28]. The reaction was quenched with 100 mM NH4CO3 for 30 min. Afterwards, the sample was dried in a speedvac, re-dissolved in 8 M Urea, reduced with 5 mM Tris(2-carboxyethyl)phosphine (TCEP) and alkylated with 10 mM iodoacetamide. For digestion, the sample was diluted to 1 M urea and trypsin (protease/protein ratio 1:50) was added over night. Digestion was stopped with 5% formic acid and peptides were purified by C18 clean-up. Dried peptides were dissolved in 20 µl 0.1% formic acid and 30% acetonitrile. Cross-linked peptides were enriched by peptide size-exclusion chromatography with Superdex Peptide PC 3.2/30 column (GE Healthcare, Uppsala) using running buffer containing 30% acetonitrile and 0.1% formic acid. SEC fractions were then dissolved in 5% acetonitrile and 0.1% formic acid, iRT peptides (Biognosys) were spiked to each sample before LC–MS/MS analysis for quality control and retention time alignment.

**Cross-linking of Dyrk2 purified from mammalian cells**. Eight subconfluent (80%) 150 mm plates of T-REx-HEK293 Flp–In cells stably expressing Strep/HA-tagged Dyrk2 were incubated for 24 h with 1 µg/ml tetracycline to induce protein expression. Cells were collected by centrifugation (400×g, 5 min) and snap-frozen. Then the cell pellet was lysed in 4 ml lysis buffer (0.5% NP40, 50 mM HEPES (pH 7.5), 150 mM NaCl, 50 mM NaF, 400 nM Na3VO4 supplemented with 1 mM PMSF, 1.2 µM Avidin, and protease inhibitor cocktail (P8849, Sigma)). The cleared cell lysate was incubated with disuccinimidyl suberate (DSS) (Sigma) cross-linked Strep-Tactin beads (IBA LifeSciences) for 1 h on a rotation shaker. Upon washing two times with lysis buffer and three times with the same buffer but without protease inhibitor and detergent, the cross-linking reaction of the affinity-purified Dyrk2 was performed by adding 1 mM isotope labeled di-succinimidylsuberate (DSS-d0, DSS-d12) (CreativeMolecules Inc.) at 37 °C for 30 min. The reaction was quenched with 100 mM NH4CO3 for 30 min and, after pelleting the beads, the supernatant was removed. Cross-linked Dyrk2 bound to the beads was denatured with 8 M urea, reduced with 5 mM Tris(2-carboxyethyl)phosphine TCEP, and alkylated with 10 mM iodoacetamide. The sample was diluted with 100 mM NH4CO3 to 4 M urea and Dyrk2 was digested on the beads with 0.5 µg LysC (Wako) for 3 h, followed by dilution to 1 M urea and digestion by 0.8 µg trypsin over night. The digestion was stopped by addition of 5% formic acid and the peptides were purified by C18 UltraMicroSpin columns and dried in a speedvac. The dried peptides were dissolved in 2% acetonitrile and 0.1% formic acid.

**Data acquisition for interaction analysis**. LC–MS/MS analysis was performed on an Orbitrap Elite mass spectrometer (AP–MS) (ThermoScientific) coupled to an Easy-nLC 1000 system (ThermoScientific) and LTQ Orbitrap XL mass spectrometer (BioID–MS) (ThermoScientific) with Xcalibur software (4.1) (Thermo) coupled to an Easy-nLC II system (Proxeon). For samples derived from affinity purification peptides were separated on a Thermo PepMap RSLC column (15 cm length, 75 µm inner diameter) with a 60 min gradient from 5 to 35% acetonitrile at a flow rate of 300 nl/min whereas for BioID samples a gradient of 90 min (5–35% acetonitrile) was used. The mass spectrometer was operated in data-dependent acquisition (DDA) mode with the following parameters: one full FTMS scan (350–1600 $m/z$) at 120,000 resolution followed by MS/MS scans on the fifteen most

abundant precursors with a charge state of +2 or higher, activation type = CID, isolation width = 1 $m/z$, normalized collision energy = 35%, activation $Q = 0.25$, activation time = 10 msec. The minimum signal threshold of precursors to induce MS/MS scans was set to 500 ion counts. For data acquisition, a dynamic exclusion for the selected ions was set: repeat count = 1, repeat duration 30 s, exclusion size list = 500, exclusion duration = 30 sec, exclusion mass width (relative to reference mass) = low 10, high = 10. For AP–MS, 10% and for BioID, 5% of the sample was injected.

**Data acquisition for total proteome analysis.** Samples for total proteome analysis were measured on a Sciex TripleTOF 6600 instrument (AB Sciex Instruments) with SCIEX Analyst v1.7 software equipped with a NanoLC Ultra 2D (Eksigent). Peptides were separated using a self packed C18 column (PicoTipTM Emitter, New Objective, Woburn, USA) (21 cm length, 75 μm inner diameter) with a 60 min gradient from 2 to 35% buffer B (98% acetonitrile and 0.1% formic acid) at a flow rate of 300 nl/min. The mass spectrometer was operated in SWATH-mode using 64 variable windows between 400 and 1200 $m/z$ (1 $m/z$ overlap). The collision energy for each window was determined based on calculation for peptides with a charge state of 2+ adding a spread of 15 eV. For total proteome analysis, 1 μg peptides of the sample was injected.

**Data acquisition for phosphoproteomic analysis.** To generate a phosphopeptide-specific assay library, 26 phosphopeptide-enriched samples were acquired in high-resolution data-dependent acquisition mode on TripleTOF 6600 (AB Sciex Instruments). Phosphopeptides were separated by liquid chromatography (NanoLC Ultra 2D, Eksigent) at 0.3 ml/min flow rate interfaced to a NanoSpray III source (AB Sciex Instruments). As column material, a PicoTipTM Emitter (75 μm inner diameter) was in-house packed with C18 beads (MAGIC, 3 μm, 200 Å, Michrom BioResources, Auburn, USA) and cut to a length of 21 cm. Phospho-peptides were separated on a 120 min long linear gradient from 5% solvent A (2% acetonitrile and 0.1% formic acid) to 35% solvent B (98% acetonitrile and 0.1% formic acid). The twenty most intense precursor ions with a charge state between +2 and +5 were selected for CID fragmentation, and were excluded for re-fragmentation for 20 s. MS1 scan time was 300 ms over a mass to charge range of 360 to 1460, followed by 20 MS2 spectra measurements with 150 ms per spectra from 50 to 2000 $m/z$. For CID fragmentation, a collision energy spread of 15 eV was dynamically adjusted.

The same phosphopeptide-enriched samples were submitted to measurements in SWATH-mode on the TripleTOF 6600 instrument. For SWATH measurements, the gradient was shortened to 90 min from 5% solvent A (2% acetonitrile and 0.1% formic acid) to 35% solvent B (98% acetonitrile and 0,1% formic acid). Data was acquired in positive ion and high-sensitivity SWATH-mode, using 100 variable windows from 400 to 1250 $m/z$ with 1 $m/z$ overlap at the upper window boarder. A measurement time of 200 ms for MS1 precursor scans and 30 ms for each fragment ion scan was set, resulting in a 3.2 s duty time per cycle.

**Data acquisition for cross-linking analysis.** LC–MS/MS (DDA mode) was performed on Orbitrap Lumos Tribrid mass spectrometer (ThermoFischer) equipped with a Thermo easy-nLC1200 liquid chromatography system (ThermoFischer). Peptides were separated using reverse phase column (Acclaim PepMap RSLC C18 column, 2.0 μm, 75 μm*250 mm) across 60 min linear gradient from 7 to 35% (buffer A: 0.1% (v/v) formic acid, 2% (v/v) acetonitrile; buffer B: 0.1% (v/v) formic acid, 98% (v/v) acetonitrile). The data acquisition mode (data-dependent acquisition) was set to perform a cycle of 3 s with high-resolution MS scan in the Orbitrap (120,000 at 400 $m/z$) and MS/MS spectra in the ion trap. Charge state lower than 3 and bigger than 7 were rejected. The dynamic exclusion window was set to 25 s. Precursors with MS signal that exceeded a threshold of 5000 were allowed to be fragmented (CID, collision energy 35%). The ion accumulation time was set to 50 ms (MS) and 100 ms (MS/MS).

LC–MS/MS (target mode) was performed on Orbitrap Lumos Tribrid mass spectrometer (ThermoFischer) equipped with a Thermo easy-nLC1200 liquid chromatography system (ThermoFischer) using the same configuration as described for DDA mode. The data acquisition mode (PRM) was set to perform a MS1 scan followed by time scheduled targeted PRM scans acquired at variable resolution (60,000 and 120,000) fragmented as in the DDA acquisition. The quadrupole isolation window for the PRM events was set to 1.4 $m/z$ units and the duration of the time scheduled windows were set to 2 min.

**Data analysis on interactomics data.** Acquired MS/MS scans were searched against the UniProtKB/Swiss-Prot protein database (10.05.2018) with the Euler-Portal (ETH in-house software) workflow using the search engines X!TANDEM Jackhammer TPP (2013.06.15.1—LabKey, Insilicos, ISB), Comet (2016.01 rev. 3) and MyriMatch v2.1.138 considering a precursor mass tolerance of 15 ppm and a fragment mass error of 0.4 Da. Tryptic peptides with a maximum number of two missed cleavages were considered for the peptide identification search. Carbami-domethyl on cysteine residues as static modification was added.

High confident interactors of AP–MS experiments were determined by SAINTexpress[53] with default parameters using spectral counts obtained from EulerPortal. Eighteen Strep/HA-GFP pulldowns processed and measured in the

same way like the samples were used as controls for SAINTexpress scoring. Proteins with a SAINT score > 0.95 were filtered additionally against control runs of the CRAPome database (http://crapome.org/;[54] 411 pulldowns). IDs identified in > 15% of the CRAPome control runs and in the SH-GFP pulldowns of this study were removed to obtain the final set of high confident interactors.

High-confidence interactors of BioID–MS experiments were determined by SAINTexpress with default parameters as described. As control nine experiments with BirA*-GFP processed and measured in the same way like the samples were considered for the SAINT scoring. Furthermore, cells expressing the bait protein but not treated with biotin (nine experiments for Dyrk2, three experiments for Ubr5, DDB1, VprBP) were used as additional control for the SAINT scoring in order to identify and remove endogenously biotinylated proteins. IDs assigned with a SAINT score = 1 were considered as high confident interactors.

For the quantification of interactome changes in the AP–MS and BioID–MS network, LFQ MS1 intensities were determined by MaxQuant analysis v1.5.2.8 using default parameters and the UniProtKB/Swiss-Prot protein database[25]. Statistical analysis was performed by customized R scripts. Briefly, LFQ MS1 intensities were bait-normalized and missing values were imputed using random sampling from a distribution based on the 5th lowest quantile. In the following, fold changes and adjusted $p$-values were calculated.

**Network visualization and GO analysis.** The general layout of protein–protein interaction networks (Figs. 2a and 6a) was generated using Cytoscape (v3.6.0)[55]. To calculate the recall rate and represent already deposited interactions, we used as a reference database the Integrated Interaction Database (IID; release 2018.05[18]). Dot plots were generated using the ProHits-viz tool (https://prohits-viz.lunenfeld.ca/index.html[56]). GO analyses (Figs. 2b, 5g; Supplementary Fig. 2a) were carried out using DAVID (https://david.ncifcrf.gov/[32]) using primarily the BP(biological process)_DIRECT annotation in the Functional annotation tool and $p$-values for GO enrichment by a modified Fisher exact test. The network shown in Fig. 6a was prepared as follows: (i) first, we filtered proteins in our PPI and phosphoproteomics data such that they satisfy the following conditions: (A) They are present in the Cancer Census list (https://cancer.sanger.ac.uk/census, download 20190410); (B) They are regulated either at the phospho level ($|\log 2FC| > 1$, peptide FDR < 0.05, and/or at the interactome level ($|\log 2FC| > 1$, adj. $p$-value < 0.05)). (ii) We mapped the known interactions among the selected proteins based on the integrated interactome database (IID v.2018-05; http://iid.ophid.utoronto.ca/). Of all interactions reported in IID, only those annotated as experimentally validated and with associated PubMed ID(s) were considered. In the graph, isolated nodes are those for which no interaction was retrieved. (iii) We performed a gene ontology analysis using DAVID (https://david.ncifcrf.gov/). Specifically, we considered the Biological Process domain (as opposed to Cellular Component or Molecular Function domains), and the default "Direct" category with default Homo sapiens background. To group GO terms, we used (i) shared parent terms identified with amiGO2 v.2.5.12 (which provides an inferred tree view of GO terms), (ii) and/or on shared assigned genes, (iii) and/or semantic similarity not captured in the amiGO2 inferred tree view.

**Data analysis on total proteome data.** DIA-SWATH data were analyzed with the OpenSWATH workflow[57] implemented on the in-house EulerPortal platform after conversion of the raw SWATH wiff files into mzXML format by ProteoWizard (3.0.8851). Spectra were extracted using the combined human assay library with a fragment ion $m/z$ extraction window of 0.05 Th and a retention time extraction window of 480 s. Detected features were aligned with TRIC with a target FDR of 0.01. If no peak group was detected, the area was re-quantified as described[58]. The OpenSWATH data output was further processed by the R package SWATH2-stats[59]. The data were filtered with a global m-score cut-off of 3.9811E−5 resulting in a protein FDR of 0.0284 using an estimated fraction of false targets (FFT) of 0.46. The resulting data matrix was further filtered for proteotypic peptides. The following quantitative and statistical analysis was carried out by mapDIA v1.2.1[60]. The filtered data matrix was normalized using total intensity normalization and an independent study design with a minimum correlation of 0.1 was chosen. For the calculation of fold changes and $p$-values, at least one peptide per protein and between 3–6 fragments per peptide were selected.

**Generation of a phosphopeptide SWATH-assay library.** The 26 in DDA mode measured files were used to generate a phosphopeptide-specific SWATH library. The raw MS files were converted to the mzXML file format with ProteoWizard (3.0.8851)[61]. For peptide identification, the data were searched using EulerPortal with the search engines X!TANDEM Jackhammer TPP (2013.06.15.1—LabKey, Insilicos, ISB), omssacl (version 2.1.9), and Comet (2016.01 rev. 3) against the human proteome (UniProtKB 10.05.2018) appended with reversed DECOY sequences for scoring. Tryptic peptides with a maximum number of two missed cleavages were considered for the peptide identification search. Carbamidomethyl on cysteine residues as static and phosphorylation on serine, threonine, and tyrosine, and oxidation on methionine residues as variable modifications were added. The maximum precursor mass error was 50 ppm and the fragment mass error was set to 0.1 Da. Identified peptide sequences were analyzed with the Trans-Proteomic Pipeline (TPP v4.7 POLAR VORTEX rev 0) applying PeptideProphet, iProphet,

and Proteinprophet scoring. The search result was filtered at 0.01 FDR corresponding to an iprophet-peptide probability of 0.8829. To assess the localization of the phosphorylation site for each annotated spectra, the same strategy as reported in ref. [62] was used. In brief, a global false localization rate (FLR) was calculated using LuciPHOr2[30], allowing the classification of localized and non-localized phosphopeptides. All phosphopeptides with a lower FLR of 0.01 were annotated as localized, and all above the FLR threshold were annotated as non-localized phosphopeptides. For all non-localized phosphopeptides, the unique UniProtein identifier was expanded with "Phospho_1". The spectral assay library was built using the Trans-Proteomic Pipeline as previously reported[63] independently for localized and non-localized phosphopeptides and rejoined before DECOY and SWATH-assay generation. The final library contains 3239 localized and non-localized phosphopeptides, which origin from 1291 unique phosphoprotein groups.

**Phospho-SWATH data quantification**. The extraction of quantitative phospho-peptide data was conducted with the OpenSWATH workflow[57] integrated in the EulerPortal tool and the Dyrk2-mutant-specific phospho-SWATH library generated for this project. The raw wiff files were converted to profile mzXML files with ProteoWizard (3.0.8851). All recorded phospho-SWATH maps were extracted with OpenSWATH applying an $m/z$ fragment ion extraction window of 0.05 Th around the expected mass of the fragment ions and a retention time window of ±300 s after iRT-alignment. Scoring of the peaks was conducted with PyProphet applying 10 different scores. Detected features were mapped between runs, and re-aligned using a regression with a local minimum spanning tree model, with a 0.01 target FDR. If there was no feature identified within a run, the peptide and fragment signals were obtained by quantification of the respective background signal at the expected elution time of the peptide, allowing to boost the completeness of the resulting data matrix[58]. Next, the obtained intensities for all fragment features from Open-SWATH were processed with SWATH2stats[59], allowing for elaborate filtering of the data matrix. Low quality features were removed with an $m$-score threshold of 0.01 and additionally a fragment feature had to be detected in at least 30% of all phospho-SWATH measurements. These conservative filtering resulted in a data matrix of 2887 quantified unique phosphopeptides from 1193 uniprot IDs across all conditions. Data were normalized using TIS (total ion spectra) normalization within the mapDIA framework[60]. Group comparison between each of the seven conditions against the wild-type Dyrk2 phospho signal was performed within mapDIA. This allowed us to obtain the log2FC values and to estimate the significance of change in intensity of phosphorylation of each given localized or non-localized phosphopeptide with a given adj. $p$-value (or FDR).

**Analysis of cross-linking data**. MS data were converted to mzXML format with msConvert and searched with xQuest/xProphet[28] against a database containing the fasta sequence of Dyrk2 and its decoy sequence. Cross-linked peptides with a minimal length of 5 amino acids and a xQuest ld (linear discriminant) score higher than 25 were considered and selected for further analysis. The selected cross-linked peptides were quantified across different condition with a targeted proteomics approach (PRM = Parallel Reaction Monitoring). To perform the targeted analysis, we generated a library with the elution time of the precursor, $m/z$ value and the charge of the cross-linked peptides (heavy and light form) identified by xQuest. Data analysis of acquired targeted cross-linked peptides were performed in Skyline v.4.1[64]. The common transition (fragment ions that can be detected in the spectra of light and heavy cross-linked peptides) were imported as described[65]. Peptides were analyzed manually, and correct identification was assigned on the basis of the following criteria: (1) retention time matching with the identification by the xQuest analysis (±2 min), (2) co-elution of heavy and light cross-linked peptides, (3) matching of the peak shape and intensity for six common transitions of heavy and light cross-linked peptides. The abundance of cross-linked peptides was analyzed by summing the integrated area of twelve transitions per peptide (respectively six transition for the heavy and light form of the cross-linked peptide). Transitions with a signal to noise ratio <5 were filtered out for the quantification. Cross-linked peptides were normalized for the intensity of two non-cross-linked Dyrk2 peptides (acquired in PRM mode) and missing value were imputed with the minimum value identified in the analysis.

**Reporting summary**. Further information on research design is available in the Nature Research Reporting Summary linked to this article.

## Data availability

The mass spectrometry proteomics data have been deposited to the ProteomeXchange Consortium via the PRIDE[66] partner repository with the data set identifier PXD015687. The source data underlying Figs. 2, 3, 4, 5a, c–f, 6, 7, 8a–c and Supplementary Figs. 1b, 3c, e, 4a, 5b–g, 6a, 7a, b are provided as source data file. All other data are available from the corresponding authors on reasonable request. Source data are provided with this paper.

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

## Acknowledgements

The work was supported by the SystemsX.ch project PhosphoNetX PPM to R.A., the Swiss National Science Foundation (Grant No. 3100A0-688 107679 to R.A.), the European Research Council (ERC 20140AdG 670821 to R.A.), the Innovative Medicines Initiative project ULTRA-DD (FP07/2007-2013, Grant No. 115766 to R.A. and M.G.). M.M. was supported by a Long-Term Fellowship from the European Molecular Biology Organization (ALTF 928-2014). We thank Marija Buljan and members of the Kutay lab for very helpful discussions and critical input.

## Author contributions

R.A., M.M., and M.G. conceived the idea and designed the experiments. M.M. performed and analyzed the affinity purification experiments. Phosphoproteomic experiments were performed and analyzed by M.M., F.F., and R.C. The cross-linking experiments were performed by M.M. The data analysis and visualization of cross-linking experiments were performed by F.U. and M.M. M.M. and A.v.D. generated cell lines and performed cloning and cell culture experiments. K.R. and M.M. performed and analyzed the proximity proteomics experiments. Immunofluorescence analysis was performed by R.C. M.M., R.A., and R.C. wrote the manuscript with inputs from all other authors. R.A. and M.G. supervised the study.

## Competing interests

R.A. holds shares of Biognosys AG, which operates in the field covered by the article. The remaining authors declare no competing interests.
