## [Peer Review File · Nature Communications]

Reviewers' comments:

Reviewer #1 (Remarks to the Author):

Mehnert et al. use a series of proteomics approaches (AP-MS, BioID-MS, crosslinking-MS, proteomics, and phosphoproteomics) to investigate the impact of mutations of the DYRK2 kinase on its interactome, topology, and downstream signaling. They describe this as a multi-layered workflow. DYRK2 was chosen because of its known participation in a multi-protein complex and possible role in cancer progression. The authors perform AP-MS and BioID-MS experiments to map the interactome of WT DYRK2 and five mutant versions and find mutation-specific changes in the interactome. They compared the phosphorylation state of WT and mutant DYRK2 and again found that different mutants affected phosphorylation and topology to different degrees, which correlated with the effects of the mutations on the interactome. Next, they investigate the topology of WT and a subset of mutants by cross-linking mass spectrometry and found some increased and decreased interactions compared to WT. It is unclear how informative these results are. Finally, the authors conduct proteomics and phosphoproteomics studies. While the proteome of cells expressing the different mutants displayed little differences to WT, the phosphoproteome was differentially affected by different mutations. To do so, the authors knock out endogenous DYRK2 and reintroduce WT and mutants. The authors employ SWATH-MS for the phosphoproteomics analysis and identify only 2888 unique phosphopeptides across all conditions which limits the impact of this study.

This is a carefully conducted study. The proteomics analyses are of high quality. The authors stress that the advantage of their approach is that multi-layered proteomics information identifies how different mutations affect signaling and helps to pinpoint underlying mechanisms that thereby discriminate mutants, even those that have a similar predicted "damage score".

The major critique of this study is that the authors do not demonstrate the significance of their findings. They observe changes, some of which are mutation-specific but do not show how they affect cellular function. This limits the impact of this study. The authors generate a knockout and overexpression system but do not describe any cellular effects of these mutants.

They conclude that mutations that strongly impact kinase function result in differences on several proteomics levels (interactome, topology, phosphoproteome). The authors should perform in vitro kinase assays of the mutants to see if their observation directly correlates with kinase activity as assumed for the kinase-dead mutant. Their multi-layered approach would be more justified if kinase activity would not be sufficient to discriminate mutations with similar damage score.

The authors identify interactions of DYRK2 with components of the nuclear pore complex as well as some altered sites on other nuclear pore complex proteins. They speculate that interaction with some components guides the phosphorylation of others. However, they do not show if nuclear transport is altered in cells expressing DYRK2 mutants.

Minor comments:

1. The authors should include in S1 a table indicating how often the respective mutations are observed.
2. Line 139-140 the authors refer to supp figure 2b, I think they mean supp figure 1b.

Reviewer #2 (Remarks to the Author):

In the present manuscript Mehnert et al. showcase a highly detailed, multi-layered proteomics study, meticulously dissecting the molecular mechanisms involved in the pathogenicity of five disease-associated mutations in the Dyrk2 protein kinase. In an extensive series of expertly conducted experiments, comprising AP-MS and BioID-MS, cross-linking MS, quantitative phosphoproteomics, the authors generate a vast amount of new and important biological knowledge on disease-associated mutations in the Dyrk2 protein kinase.

The -in some parts lengthy and verbose- manuscript is well written, the authors' conclusions are all supported by the data presented and I endorse its publication in its present form.

Reviewer #3 (Remarks to the Author):

In this manuscript the authors study the effect of cancer related mutations in the Dyrk2 protein kinase using several proteomic approaches. The authors selected five cancer related mutations of Dyrk2 and created stable cells with these mutations to then compare wild type interactions with mutant interactions using affinity purification coupled to mass spectrometry and proximity labeling using BioID. Next the authors used cross linking approaches to analyze potential changes in the structure of Dyrk2 wild type and mutant proteins. This was followed by a phosphoproteome analysis of cancer mutations in cells. All of these methods are widely used in the research community. Lastly, the authors attempted to integrate this information into a network of changes induced by mutations in Dyrk2. Overall, this manuscript has a number of major issues at this time.

The first major issue is while there are multiple proteomics experiments conducted here and an attempt is made to map these datasets into a network model, the manuscript does not provide novel and actionable insights into Dyrk2 biology, the function of this protein kinase, and how to use this information to better understand cancer biology. Figure 6 is emblematic of this issue where the authors attempt to combine the datasets into functional insights, but no validation of any insights are provided. It is also unclear how information was chosen to be included in this figure other than a qualitative analysis of other datasets. A more sophisticated analysis would include a mathematical and statistical justification for inclusion of data in this figure and a way to then use this figure to make data driven decisions for pursuing additional experiments.

A second major issue somewhat related to the first issue is the statistical and mathematical decision making in the manuscript needs better explanation and justification. Each individual proteomic experiment must be supported by at least three biological replicates and the statistically supported data then described and used in each figure. As it is written, the statistical analysis of the AP-MS and BioID is provided, but this appears to be lacking for the phosphoproteomics and cross linking MS datasets. Given the cumulative nature of the data and the importance of assembling a statistically supported network, it is critical that each proteomic experiment be conducted and analyzed with sufficient rigor in order to have a rigorously supported final network.

A third major issue is the cross linking experiment is of limited value. The authors conducted this experiment on purified proteins from insect cells, which only provides information on Dyrk2 itself. This does not provide any insights regarding how these cancer mutations are affecting interactions of Dyrk2. The authors should carry out a study where AP-MS isolated Dyrk2, Dyrk2 KR, and Dyrk2 RL are then treated with cross linking reagents to determine the likely changes in interaction interfaces induced by these mutants. This type of information would be far more valuable to integrate with the other datasets in the manuscript than the limited insights provided by cross linking of the Dyrk2 proteins alone.

Response to Reviewers

We would like to thank all reviewers for the helpful and constructive comments about our work. In the following we provide first a summary of experiments that were carried out and are added to the manuscript text and figures. Then we address all other points raised by the reviewers in separate comments. Changes in the manuscript are marked in yellow and we provide information about the corresponding Page and Line in the comment section.

Summary of the performed experiments and new results

As requested by the editor and reviewers we performed several new experiments to confirm and validate the biological significance of our findings and to gain further insights into the functional role of Dyrk2 in cancer related processes.

First, we performed colony formation and cell proliferation assays with Dyrk2 KO cells and cells expressing kinase mutants to relate the observed proteomic patterns to cellular phenotypes. Indeed, we observed a significant effect of Dyrk2 KO cells on the ability to form colonies and on the proliferation rate (**Figure 6b and Supplementary Figure 6a**).

Second, in order to further investigate the effect of Dyrk2 and its mutants on cellular level we overexpressed Dyrk2 WT, the kinase dead mutant Dyrk2 KR and the cancer related truncated Dyrk2 SX mutant in HeLa cells and performed cell cycle and apoptosis assays. The overexpression of Dyrk2 KR and Dyrk2 SX that showed significant changes on different proteomic layers also caused cellular phenotypes reflected by shifts in cell cycle phases and the apoptosis rate (**Figure 6c and Supplementary Figure 6b**).

Third, we performed an *in vitro* phosphorylation assay to validate the nuclear pore subunit NUP214, a putative docking site for nuclear transport processes [1], as new direct substrate of Dyrk2.

Fourth, since it was recently shown that the assembly of nuclear pore subunits is regulated by their phosphorylation state [2] we carried out microscopy studies to investigate the effect of Dyrk2 dependent phosphorylation on the association of NUP214 with the nuclear pore complex. We observed subtle changes in the localization of NUP214 at the nuclear envelope (**Figure 1 Response to Reviewers**) consistent with an putative involvement of Dyrk2 in nuclear pore assembly.

Fifth, we performed *in vitro* kinase assays with recombinant Dyrk2 variants purified from SF9 insect cells and demonstrated that the kinase activity of the Dyrk2 mutants used in this study with the exception of the kinase dead mutant (Dyrk2 KR) is not affected indicating that the observed proteomic phenotypes are not directly correlated with kinase activity (**Supplementary Figure 5h**).

Finally, we provide a more extensive description of our data production, filtering and combination strategy used for generating the network model shown in **Figure 6a**. In this regard, we added further information in the main text and in the Material and Method section. Please see more details on these and other introduced changes in the manuscript text (marked in yellow) and in our individual replies to the reviewers below. Overall, the results from the extensive new experiments validated the results and were consistent with the statements in the initial version of the paper and thus made the paper stronger.

Reviewer #1

Mehnert et al. use a series of proteomics approaches (AP-MS, BioID-MS, crosslinking-MS, proteomics, and phosphoproteomics) to investigate the impact of mutations of the DYRK2 kinase on its interactome, topology, and downstream signaling. They describe this as a multi-layered workflow. DYRK2 was chosen because of its known participation in a multi-protein complex and possible role in cancer progression. The authors perform AP-MS and BioID-MS experiments to map the interactome of WT DYRK2 and five mutant versions and find mutation-specific changes in the interactome. They compared the phosphorylation state of WT and mutant DYRK2 and again found that different mutants affected phosphorylation and topology to different degrees, which correlated with the effects of the mutations on the interactome. Next, they investigate the topology of WT and a subset of mutants by cross-linking mass spectrometry and found some increased and decreased interactions compared to WT. It is unclear how informative these results are. Finally, the authors conduct proteomics and phosphoproteomics studies. While the proteome of cells expressing the different mutants displayed little differences to WT, the phosphoproteome was differentially affected by different mutations. To do so, the authors knock out endogenous DYRK2 and reintroduce WT and mutants. The authors employ SWATH-MS for the phosphoproteomics analysis and identify only 2888 unique phosphopeptides across all conditions which limits the impact of this study.

This is a carefully conducted study. The proteomics analyses are of high quality. The authors stress that the advantage of their approach is that multi-layered proteomics information identifies how different mutations affect signaling and helps to pinpoint underlying mechanisms that thereby discriminate mutants, even those that have a similar predicted “damage score”.

Point 1:

The major critique of this study is that the authors do not demonstrate the significance of their findings. They observe changes, some of which are mutation-specific but do not show how they affect cellular function. This limits the impact of this study. The authors generate a

knockout and overexpression system but do not describe any cellular effects of these mutants.

Author reply:

We thank reviewer #1 for this comment and we acknowledge that the investigation of cellular effects related to the kinase mutants used in this study is crucial to understand and demonstrate the biological significance of the findings derived from the proteomic workflow. In order to address this question we performed the series of cellular assays described above (colony formation, proliferation and apoptosis assays) (Material and Methods, **Page 20-21, Line 738-763**) with the Dyrk2 KO cell line and different Dyrk2 mutants. Here, we mainly focused on mutants that showed a strong phenotype on different proteomic layers. The phosphoproteomic and interaction data in the original paper show that certain Dyrk2 mutants (e.g. Dyrk2 KR and Dyrk2 SX) affect the binding and the phosphorylation state of known cancer driver proteins such as TP53, BCLAF1 and CXCR4 or the transcriptional regulator protein Mep50 (**Figure 6a** and **Supplementary Table 1**) which play crucial roles in cell proliferation and cell cycle regulation. We therefore first analysed the effect of the deletion of Dyrk2 on cell proliferation by a colony formation assay which is based on the ability of single cells to proliferate without cell contact and to form colonies, thereby representing evidence of aberrant and strongly dysregulated cell growth. Additionally, we estimated the cell growth rate using an MTT assay which is routinely used to measure cell proliferation. This assay is based on the measurement of a dye which is produced by an enzymatic reaction only in living cells and is therefore used to quantify cell viability or cell proliferation upon different kind of perturbations.

In the colony formation assay we observed a significantly increased rate of colony formation for Dyrk2 KO cells (**Figure 6b**), indicating an increase in cell proliferation which could be confirmed by the MTT assay (**Supplementary Figure 6a**). This finding is in line with a previous study showing that stable knockdown of Dyrk2 elevates cell proliferation in MCF-7 cells and tumor growth in xenograft mice studies using HeLa cells transiently transfected with Dyrk2 siRNA [3]. The new results thus support the proposed tumor suppressor function of Dyrk2.

We then repeated the MTT assay with a selection of Dyrk2 mutants (Dyrk2 KR and Dyrk2 SX) that showed either a strong or mild effect on different proteomic layers. In contrast to Dyrk2 KO cells the tested Dyrk2 mutants show no significant effect on cell proliferation (data not shown). However, cell cycle analysis revealed mild but significant shifts in cell cycle phases for both Dyrk2 KR and the cancer related Dyrk2 SX mutant (**Supplementary Figure 6b**). Interestingly, the expression of Dyrk2 SX leads to an increased fraction of apoptotic cells detected in the cell cycle analysis of HeLa CCL2 cells. This finding was further confirmed by

an Annexin V-FITC apoptosis assay (**Figure 6c**) that we additionally performed with this mutant in another cancer cell line (MDA-MB-231 cells). During apoptosis phosphatidylserine molecules are translocated to the outside of the cell membrane and can be stained by Annexin V-FITC. The fraction of Annexin V-FITC stained cells (apoptotic cells) is then measured by flow cytometry.

Dyrk2 is suggested to be directly involved in DNA damage and promoting apoptosis by phosphorylating p53 [4]. Indeed, both the KR and SX mutants affect phosphorylation and interaction to proteins such as p53 or CXCR4 that are involved in apoptotic processes [5]. Interestingly, Dyrk2 KR and Dyrk2 SX showed opposite effects on the apoptosis rate (**Figure 6c and Supplementary Figure 6b**). The inactivation of Dyrk2 by the KR mutation inhibits or reduces apoptosis in keeping with previous studies [6], whereas the expression of Dyrk2 SX results in the opposite effect and increases apoptosis. This might be explained by the fact that in comparison to Dyrk2 WT or other tested mutants Dyrk2 SX shows elevated protein levels (**Supplementary Figure 5c**) which may negatively influence cell viability. The fact that routinely used MTT assays were not able to detect significant proliferation changes between the different mutants indicates that such assays are not necessarily sufficient to detect mild cellular phenotypes and to characterize protein point mutations. The results therefore suggest that the multi-layer proteomic analysis is more sensitive in detecting perturbed cellular processes than proliferation assays.

Together, by applying a series of cellular assays such as colony formation, proliferation and apoptosis assays we could detect cellular effects of certain mutants used in this study and relate them to their proteomic phenotypes. Specifically, we observed cellular phenotypes for those mutants that also showed significant changes on different proteomic layers. We added the new information to the results section of the manuscript text (**Page 13, Line 458 - 471**).

Point 2:

They conclude that mutations that strongly impact kinase function result in differences on several proteomics levels (interactome, topology, phosphoproteome). The authors should perform in vitro kinase assays of the mutants to see if their observation directly correlates with kinase activity as assumed for the kinase-dead mutant. Their multi-layered approach would be more justified if kinase activity would not be sufficient to discriminate mutations with similar damage score.

Author reply:

We thank reviewer #1 for this valuable comment and followed the suggestion to perform *in vitro* kinase assays for the Dyrk2 mutants used in this study to determine whether the observed mutant specific phenotypes are directly related to kinase activity. For this we

expressed all mutants used in the study in SF9 insect cells and affinity-purified them using an engineered FLAG tag. As already discussed in the main text of the manuscript Dyrk2 SX could not be purified possibly due to a misfolding of this mutant in SF9 cells. The purified Dyrk2 mutants were then tested for their catalytic activity using the commercial ADP-Glo *in vitro* kinase assay kit (Material and Methods, **Page 21-22, Line 771 - 781**). Briefly, after the kinase reaction ADP-Glo reagent is added to terminate the reaction and to deplete remaining ATP. Then the kinase detection reagent is added which converts the produced ADP to ATP that is detected by a luciferase reaction. The luminescence signal correlates with the amount of ADP that was generated by the kinase reaction and is therefore indicative of the kinase activity. In our experimental setup the ATP/ADP conversion was measured for different amounts of the respective Dyrk2 variant in a given time (1h). For Dyrk2 WT we observe a kinase amount dependent exponential increase of the ATP/ADP conversion followed by a saturation of the reaction (**Supplementary Figure 5h**). Dyrk2 PL, SP and RL show a very similar increase in ATP/ADP conversion as Dyrk2 WT indicating that the kinase activity is not affected by these point mutations. Dyrk2 SL shows a slightly reduced response to the added Dyrk2 amount for the first two measurement points which is most likely due to experimental inaccuracies, specifically in pipetting small volumes. In contrast, the kinase dead mutant shows, as expected, a clearly reduced rate of ATP/ADP conversion for the same kinase amounts indicating a strongly reduced activity. The same reduction in activity is observed for Dyrk2 WT treated with the Dyrk kinase inhibitor harmine which serves as internal control and demonstrates the functionality of the assay. Overall, the Dyrk2 variants tested showed two activity patterns based on ATP/ADP conversion as a function of the Dyrk2 amount added. The first pattern includes mutants that show Dyrk2 WT-like activity such as Dyrk2 SP, PL, RL and SL while the second pattern represents mutants with reduced kinase activity such as the kinase dead mutant (Dyrk2 KR) and Dyrk2 WT treated with the kinase inhibitor harmine. In conclusion, we could confirm that with the exception of the Dyrk2 KR mutant the other kinase mutants are still catalytically active and show comparable specific activity like Dyrk2 WT. Also the activity of the Dyrk2 RL mutant which bears a mutation close to the activation loop is not altered compared to Dyrk2 WT, a finding that was already suggested from the phosphoproteome analysis (**Figure 5a**). This indicates that phenotypes observed for Dyrk2 RL, in particular on interactome level, potentially result from topological changes supported by the topological XL-MS analysis as opposed to changes in kinase activity (**Figure 4f**). Together, we show that the observed phenotypes do not necessarily correlate with kinase activity but can also be due to changes in protein- protein interactions of the mutants. Further, this result demonstrates that information from our workflow is useful to discriminate cancer mutations with similar damage probability score which cannot be easily achieved by

other approaches. We added the new information to the results section of the manuscript text (**Page 10, Line 363 - 366**).

Point 3

The authors identify interactions of DYRK2 with components of the nuclear pore complex as well as some altered sites on other nuclear pore complex proteins. They speculate that interaction with some components guides the phosphorylation of others. However, they do not show if nuclear transport is altered in cells expressing DYRK2 mutants.

Author reply:

Our interaction analysis identified the entire nuclear Y-complex as new interaction partner of Dyrk2 (**Figure 2a**). The nuclear Y-complex consists of several subunits forming a Y-shaped structure that is stably anchored within the nuclear pore complex (NPC) and serves as a structural scaffold of the complex [7]. Phosphoproteomic data further indicate that Dyrk2 affects phosphorylation of the subunits NUP214 and NUP88 (**Supplementary Figure 6c**) that are direct binding partners of the Y-complex [7, 8]. Interestingly, certain cancer related Dyrk2 mutants perturb the interaction with the Y-complex and the phosphorylation of the associated NUP214/NUP88 subunits (**Figure 3e**). In order to elucidate a putative functional role of Dyrk2 at the NPC we first aimed to validate the Dyrk2 dependent phosphorylation of NUP214. For this we performed an *in vitro* phosphorylation assay with recombinant Dyrk2 and NUP214 (Material and Methods, **Page 21, Line 765 - 769**). We observe a clear phosphorylation of NUP214 in presence of ATP and Dyrk2 whereas without ATP the phosphorylation reaction was impeded (**Figure 6d**) confirming NUP214 as new direct phosphorylation substrate of Dyrk2. The phosphorylation of NUPs during mitosis has been shown to be important for the assembly state of the NPC [2] which influences many NPC related functions such as nuclear transport processes. Recently, the kinases PLK1 and CDK1 were identified to mediate the hyperphosphorylation of interconnecting NUPs such as NUP53 and NUP98. Notably, CDK1 (pS/pT-P) phosphorylates sites [2] with similar motif as Dyrk2 (R/K-x-x-x-pS/pT-P) [9] suggesting that Dyrk2 might represent a further kinase that mediates the hyperphosphorylation of NUPs and thus contributes to NPC disassembly (manuscript text, **Page 14, Line 494 - 503**). Additionally, also the Dyrk2 substrate NUP214 was reported to be hyperphosphorylated during mitosis [10, 11]. We therefore examined if phosphorylation of NUP214 by Dyrk2 affects the association of NUP214 with the NPC. For this we expressed Dyrk2 WT as well as catalytically inactive Dyrk2 KR and Dyrk2 SX in HeLa cells and analysed their effect on the localization of NUP214 at the nuclear envelope (NE) by immunofluorescence. Compared to Dyrk2 WT the catalytically inactive KR mutant shows a slightly increased localization of NUP214 at the NE (**Figure1 Response to Reviewers**, left

panel) indicating that the Dyrk2 driven phosphorylation of NUP214 affects the association of NUP214 with the NPC. Accordingly, the expression of the cancer related Dyrk2 SX that inhibits phosphorylation of NUP214 also results in a slightly increased targeting of NUP214 to the NE (**Figure1 Response to Reviewers**, right panel) which may, in combination with other factors, perturb NPC disassembly. We realize that the observed effect on the NUP214 localization is only subtle. Of note, this effect is comparable to the effect of previously described and published NUP98 single point mutations where it was shown that only simultaneous mutation of several phosphorylation sites in NUPs cause clear disassembly of the NPC [2]. Therefore, a strong effect of Dyrk2 alone on NPC disassembly or nuclear transport processes is not expected. However, since the observed effect of the Dyrk2 mutants is indeed very subtle and since we believe that further validation by orthogonal experiments will be required to conclusively support it, we decided not to include it in the present manuscript.

Figure 1 Response to Reviewers: Immunofluorescence analysis for the localization of NUP214 at the nuclear envelope in HeLa CCL2 cells overexpressing Dyrk2 wt or Dyrk2 KR and Dyrk2 SX, respectively. Quantification of NUP214 intensity across the nuclear envelope, $n \geq 29$ cells per condition.

Method information for NUP214 immunofluorescence analysis:

For the analysis of NUP214 localization at the nuclear envelope HeLa CCL2 cells were transfected with Strep/HA-tagged Dyrk2 variants. After 24h cells were washed and fixed on coverslips as described above. After permeabilization with Triton X-100 and blocking with BSA cells were stained for NUP214 (anti-NUP214 (1:1000), ab70497, Abcam) and the HA-epitope (HA.11 (1:1000),901513, BioLegend) for 1h. After washing with PBS cells were stained with fluorescent secondary antibodies (IgG Alexa Fluor 488, A11034, Thermofisher) and (IgG Cy5, A10524, Thermofisher). Confocal microscopy was carried out on an Olympus FluoView 3000 using a 60x1.35 NA, Oil UPlanSApo objective. 61, 47, and 49 images were acquired for Dyrk2 WT, SX, KR, respectively. Images were analyzed in Imagej/Fiji (v2.0.0-rc-69/1.52p) and RStudio (v.1.2.5001) as follows: intensity profiles of NUP214 were

measured with a 5 pixel thick line at a nuclear diameter of about 250 pixel, with 60 pixel wide cytoplasmic overhangs on both sides of the rim. The average of all recorded profiles (33, 29 and 29 for WT, KR and SX, respectively) were used for comparison between Dyrk2 WT and mutants.

Minor comments:

1. *The authors should include in S1 a table indicating how often the respective mutations are observed.*

Author reply:

We thank the reviewer for the comment and we included the requested information in Supplementary Table 4.

2. *Line 139-140 the authors refer to supp figure 2b, I think they mean supp figure 1b.*

Author reply:

We corrected the respective reference.

Reviewer #2

In the present manuscript Mehnert et al. showcase a highly detailed, multi-layered proteomics study, meticulously dissecting the molecular mechanisms involved in the pathogenicity of five disease-associated mutations in the Dyrk2 protein kinase. In an extensive series of expertly conducted experiments, comprising AP-MS and BioID-MS, cross-linking MS, quantitative phosphoproteomics, the authors generate a vast amount of new and important biological knowledge on disease-associated mutations in the Dyrk2 protein kinase.

The -in some parts lengthy and verbose- manuscript is well written, the authors' conclusions are all supported by the data presented and I endorse its publication in its present form.

Author reply:

We thank reviewer #2 for the comments and for appreciating the quality of the data and the relevance of our findings.

Reviewer #3

In this manuscript the authors study the effect of cancer related mutations in the Dyrk2 protein kinase using several proteomic approaches. The authors selected five cancer related mutations of Dyrk2 and created stable cells with these mutations to then compare wild type interactions with mutant interactions using affinity purification coupled to mass spectrometry and proximity labeling using BioID. Next the authors used cross linking approaches to analyze potential changes in the structure of Dyrk2 wild type and mutant proteins. This was followed by a phosphoproteome analysis of cancer mutations in cells. All of these methods are widely used in the research community. Lastly, the authors attempted to integrate this information into a network of changes induced by mutations in Dyrk2. Overall, this manuscript has a number of major issues at this time.

Point 1:

The first major issue is while there are multiple proteomics experiments conducted here and an attempt is made to map these datasets into a network model, the manuscript does not provide novel and actionable insights into Dyrk2 biology, the function of this protein kinase, and how to use this information to better understand cancer biology. Figure 6 is emblematic of this issue where the authors attempt to combine the datasets into functional insights, but no validation of any insights are provided. It is also unclear how information was chosen to be included in this figure other than a qualitative analysis of other datasets. A more sophisticated analysis would include a mathematical and statistical justification for inclusion of data in this figure and a way to then use this figure to make data driven decisions for pursuing additional experiments.

Point 2:

A second major issue somewhat related to the first issue is the statistical and mathematical decision making in the manuscript needs better explanation and justification. Each individual proteomic experiment must be supported by at least three biological replicates and the statistically supported data then described and used in each figure. As it is written, the statistical analysis of the AP-MS and BioID is provided, but this appears to be lacking for the phosphoproteomics and cross linking MS datasets. Given the cumulative nature of the data and the importance of assembling a statistically supported network, it is critical that each proteomic experiment be conducted and analyzed with sufficient rigor in order to have a rigorously supported final network.

Author reply:

We thank Reviewer #3 for these two important comments, and we will address them jointly, since they broadly pertain the same theme.

As of our experiments - AP-MS, BioID, proteome and phosphoproteome profiling, as well as cross-linking – all have been carried out in biological triplicates and they have all been evaluated using widely utilised tools in the field, including accepted statistical tools and practices. We acknowledge that our choices may have been poorly explained, or not explicitly/consistently mentioned in the text or the figures. For this reason, we changed the manuscript as follows:

- Introduced in the text explicit mention of the thresholds, CVs, and other scores used.
- Indicated significance thresholds in the Figure legends.

Specifically:

In general adj. p-values were calculated using the Benjamini-Hochberg correction.

- For proteome analyses: we performed a total proteome analysis by DIA/SWATH-MS, and consistently measured and quantified 5138 proteins (peptide CV over triplicates < 11%; peptide FDR < 1%) across the cell lines. Only 0.2% (10 proteins, adj. p-value ≤ 0.05 , $|\log_2FC| > 1$) of proteins showed altered abundance.

- For phosphoproteome analyses: we performed a total phosphoproteome analysis by DIA/SWATH-MS with a peptide CV <16% over triplicates, we identified 2888 peptides with a peptide FDR < 1%; using LuciPhor2, which is based on a target-decoy filtering strategy that uses mass accuracy and peak intensities for site localization scoring [12], we could assign a phosphosite localisation to 2040 peptides with a false localisation rate (FLR) < 1%.

Differentially abundant phosphopeptides were defined as satisfying the following criteria: FLR < 1%, adj. p-value ≤ 0.05 , $|\log_2FC| > 1$.

- For cross-linking: the identification and statistical analysis of cross-linked peptides was performed with xQuest/xProphet [13, 14].

Specifically, only cross-linked peptides with a minimal length of 5 amino acids and an xQuest Id score higher than 25 were considered and selected for further analysis. The Id (linear discriminant) score is based on a target-decoy model and defines the separation of true and false positive cross-linked peptides [13]. Next, the selected cross-linked peptides were quantified across different conditions with a targeted proteomics approach (PRM = parallel reaction monitoring). The data analysis of acquired targeted cross-linked peptides was performed in Skyline v.4 [15]. Here, the selected cross-linked peptides were analysed manually, and correct identification was assigned on the basis of the following criteria: (1) retention time matching with the identification by the xQuest analysis (± 2 min), (2) co-elution

of heavy and light cross-linked peptides, (3) matching of the peak shape and intensity for six common transitions of heavy and light cross-linked peptides. The abundance of cross-linked peptides was determined by summing the integrated area of twelve transitions per peptide (respectively six transition for the heavy and light form of the cross-linked peptide). Transitions with a signal to noise ratio less than 5 were filtered out for the quantification. Cross-linked peptides were normalized for the intensity of two non-cross-linked Dyrk2 peptides (acquired in PRM mode) and missing value were imputed with the minimum value identified in the analysis. Please see also a graphical explanation of the data analysis strategy in **Supplementary Figure 4d**.

A detailed description of the used data analysis strategy for all proteomic layers can be found in the Material and Method section of the manuscript (**Page 27-31, Line 994 - 1139**).

- Utilized thresholds are indicated in the Figure captions. Statistically significant changes in the Dyrk2 phosphoproteome and XL data are reported as arrows in the **Figure 4a, 4f** and **Supplementary Figure 6b**, respectively. In **Figure 5a** and **Figure 5e** only those peptides or proteins that have been detected at least once as statistically significantly changed (adj. p-value ≤ 0.01 , $|\log_2FC| > 0.5$) are displayed.

As of the Cytoscape graph in **Figure 6a**, we acknowledge again that the description in both the text and the Material and Methods omitted important details, and we provide now a more exhaustive description in both sections to clarify the construction of the graph (Manuscript text: **Page 13, Line 445 – 452**; Material and Methods: **Page 28, Line 1028 - 1050**).

Specifically, the graph has been generated following these steps:

(i) First, we filtered proteins in our PPI and phosphoproteomics data such that they satisfy the following conditions: A). They are present in the Cancer Census list (<https://cancer.sanger.ac.uk/census>, download 20190410); B). They are regulated either at the phospho level ($|\log_2FC| > 1$, adj. p-value ≤ 0.05) and/or at the interactome level ($|\log_2FC| > 1$, adj. p-value < 0.05).

(ii) We mapped the known interactions among the selected proteins based on the integrated interactome database (IID v.2018-05; <http://iid.ophid.utoronto.ca/>) [16]. Of all interactions reported in IID, only those annotated as experimentally validated and with associated PubMed ID(s) were considered. In the graph, isolated nodes are those for which no interaction was retrieved.

(iii) We performed a gene ontology (GO) analysis using DAVID (<https://david.ncifcrf.gov/>). Specifically, we considered the Biological Process (BP) domain (as opposed to Cellular Component or Molecular Function domains), and the default 'Direct' category and default

background (*Homo sapiens*). GO terms that possess three properties make an objective selection of terms difficult for visualisation in a graph: first, some genes are annotated with different terms, but only one or two can be used for grouping in a graph; second, many terms are semantically overlapping, i.e. partially redundant; third, some of the terms are extremely broad and are therefore hardly informative. For instance, p53 is associated with 18/36 GO BP terms relating to our graphs ($p\text{-value} < 0.05$), and not all of them can be displayed; furthermore, several of these terms both refer to the same process and are related to one another (e.g. DNA-damage related terms: GO:2001022, GO:0006974, GO:0006284). We therefore had to select and group terms for the network represented in Figure 6a. To do that, we used the following criteria: terms were grouped if (i) they shared a (semantically related) parent term in “inferred tree view” in AmiGO2 v.2.5.12, which provides an inferred hierarchical view of GO terms; and/or (ii) they shared assigned genes (for instance, even though “base-excision repair” and “cellular response to UV” share only very generic parents, they are associated with the same genes in our network and they were grouped under the “DNA damage” label); and/or (iii) they shared clearly semantically related labels (e.g. GO:0008285~negative regulation of cell proliferation was grouped together with GO:0048147~negative regulation of fibroblast proliferation and GO:0008283~cell proliferation). By this means, we could represent approximately 60% of the GO terms retrieved by DAVID in the graph.

In summary, the graph was generated using a generalisable routine that resorts to standard bioinformatic tools and databases. We followed specific criteria for grouping the GO terms and we attempted to represent as many as possible, but some terms had to be left out of the graph because of inherent representational constraints (i.e. multiple nodes cannot be simultaneously grouped into multiple subnetworks). In this respect, we could not entirely remove a subjective component in the construction of the network. In this context, two more points are worth mentioning. First, even for very large-scale studies protein annotation needs to be manually curated/merged (e.g. protein localisation; [17]). Second, while a more stringent application of GO terms merging/selection criteria can work well on large-scale dataset (as we have done recently, Buljan M. et al., in revision), for small-scale studies domain knowledge can be leveraged to counterbalance the noise associated with GO annotation and provide more insightful information.

Point 3

A third major issue is the cross linking experiment is of limited value. The authors conducted this experiment on purified proteins from insect cells, which only provides information on Dyrk2 itself. This does not provide any insights regarding how these cancer mutations are

affecting interactions of Dyrk2. The authors should carry out a study where AP-MS isolated Dyrk2, Dyrk2 KR, and Dyrk2 RL are then treated with cross linking reagents to determine the likely changes in interaction interfaces induced by these mutants. This type of information would be far more valuable to integrate with the other datasets in the manuscript than the limited insights provided by cross linking of the Dyrk2 proteins alone.

Author reply:

We agree that XL-MS data from AP purified complexes of the tested DYRK2 mutants would be informative. In this case, our reply is twofold. First, it must be noted that our idea of performing the cross-linking experiments is related to the Dyrk2 phosphorylation changes we observe across mutants. We posited that such changes would induce changes in the protein tertiary structure, which would in turn potentially affect the interaction with other partners (**Figure 4a**). From this point of view, we would consider a XL experiment on the purified protein adequate to address the initial question we have posed. Second, we acknowledge that a cross-linking experiment on pulldowns would represent an excellent platform to validate our differential interactome data. We attempted to carry out this experiment several times, but we don't seem to be able to capture inter-protein cross-links. This is possibly due to the very low expression levels of Dyrk2 in mammalian cells and the general challenges of generating inter-protein cross linking data from affinity purified complexes which is illustrated by the fact that to date only few such studies have been published. However, we were able to observe intra-protein crosslinks for Dyrk2 at positions that could be confirmed by our crosslinking data derived from purified Dyrk2 and shown in **Figure 4**. We included these data in the **Supplementary Table 3** and use them as a useful comparison/benchmark for the other XL experiment.

References

1. Fichtman, B., et al., *Pathogenic Variants in NUP214 Cause "Plugged" Nuclear Pore Channels and Acute Febrile Encephalopathy*. Am J Hum Genet, 2019. **105**(1): p. 48-64.
2. Linder, M.I., et al., *Mitotic Disassembly of Nuclear Pore Complexes Involves CDK1- and PLK1-Mediated Phosphorylation of Key Interconnecting Nucleoporins*. Dev Cell, 2017. **43**(2): p. 141-156 e7.
3. Taira, N., et al., *DYRK2 priming phosphorylation of c-Jun and c-Myc modulates cell cycle progression in human cancer cells*. J Clin Invest, 2012. **122**(3): p. 859-72.
4. Taira, N., et al., *DYRK2 is targeted to the nucleus and controls p53 via Ser46 phosphorylation in the apoptotic response to DNA damage*. Mol Cell, 2007. **25**(5): p. 725-38.
5. Kremer, K.N., et al., *CXCR4 chemokine receptor signaling induces apoptosis in acute myeloid leukemia cells via regulation of the Bcl-2 family members Bcl-XL, Noxa, and Bak*. J Biol Chem, 2013. **288**(32): p. 22899-914.

6. Morrugares, R., et al., *Phosphorylation-dependent regulation of the NOTCH1 intracellular domain by dual-specificity tyrosine-regulated kinase 2*. Cell Mol Life Sci, 2019.
7. Bui, K.H., et al., *Integrated structural analysis of the human nuclear pore complex scaffold*. Cell, 2013. **155**(6): p. 1233-43.
8. Bastos, R., et al., *Nup84, a novel nucleoporin that is associated with CAN/Nup214 on the cytoplasmic face of the nuclear pore complex*. J Cell Biol, 1997. **137**(5): p. 989-1000.
9. Campbell, L.E. and C.G. Proud, *Differing substrate specificities of members of the DYRK family of arginine-directed protein kinases*. FEBS Lett, 2002. **510**(1-2): p. 31-6.
10. Olsen, J.V., et al., *Quantitative phosphoproteomics reveals widespread full phosphorylation site occupancy during mitosis*. Sci Signal, 2010. **3**(104): p. ra3.
11. Dephoure, N., et al., *A quantitative atlas of mitotic phosphorylation*. Proc Natl Acad Sci U S A, 2008. **105**(31): p. 10762-7.
12. Fermin, D., et al., *LuciPHOr2: site localization of generic post-translational modifications from tandem mass spectrometry data*. Bioinformatics, 2015. **31**(7): p. 1141-3.
13. Walzthoeni, T., et al., *False discovery rate estimation for cross-linked peptides identified by mass spectrometry*. Nat Methods, 2012. **9**(9): p. 901-3.
14. Rinner, O., et al., *Identification of cross-linked peptides from large sequence databases*. Nat Methods, 2008. **5**(4): p. 315-8.
15. MacLean, B., et al., *Skyline: an open source document editor for creating and analyzing targeted proteomics experiments*. Bioinformatics, 2010. **26**(7): p. 966-8.
16. Kotlyar, M., et al., *IID 2018 update: context-specific physical protein-protein interactions in human, model organisms and domesticated species*. Nucleic Acids Res, 2019. **47**(D1): p. D581-D589.
17. Huttlin, E.L., et al., *The BioPlex Network: A Systematic Exploration of the Human Interactome*. Cell, 2015. **162**(2): p. 425-440.

REVIEWERS' COMMENTS:

Reviewer #1 (Remarks to the Author):

The authors have addressed my concerns. I support the publication of the manuscript.